# Regression-adjusted Monte Carlo Estimators for Shapley Values and Probabilistic Values

**R. Teal Witter**
Claremont McKenna College
rtealwitter@cmc.edu

**Yurong Liu**
New York University
yurong.liu@nyu.edu

**Christopher Musco**
New York University
cmusco@nyu.edu

## Abstract

With origins in game theory, probabilistic values like Shapley values, Banzhaf values, and semi-values have emerged as a central tool in explainable AI. They are used for feature attribution, data attribution, data valuation, and more. Since all of these values require exponential time to compute exactly, research has focused on efficient approximation methods using two techniques: Monte Carlo sampling and linear regression formulations. In this work, we present a new way of combining both of these techniques. Our approach is more flexible than prior algorithms, allowing for linear regression to be replaced with any function family whose probabilistic values can be computed efficiently. This allows us to harness the accuracy of tree-based models like XGBoost, while still producing unbiased estimates. From experiments across eight datasets, we find that our methods give state-of-the-art performance for estimating probabilistic values. For Shapley values, the error of our methods can be $6.5\times$ lower than Permutation SHAP (the most popular Monte Carlo method), $3.8\times$ lower than Kernel SHAP (the most popular linear regression method), and $2.6\times$ lower than Leverage SHAP (the prior state-of-the-art Shapley value estimator). For more general probabilistic values, we can obtain error $215\times$ lower than the best estimator from prior work.

## 1 Introduction

As AI becomes more prevalent across health care, education, finance, and the legal system, underlying algorithmic mechanisms are growing increasingly complex. Sophisticated computational models frequently make decisions that are opaque and challenging to comprehend. This is unacceptable in contexts where decisions can have profound consequences for individuals: the ability to clearly understand and explain how an algorithmic system reaches its conclusions is paramount.

One tool that has arisen to address the challenge of understanding model behavior are *probabilistic values*, which include Shapley values, Banzhaf values, and semi-values as special cases [SK10, LL17, LEC+20, WJ23]. Originating from game theory [Sha51], probabilistic values quantify the contribution of a player by measuring how its addition to a set of other players changes the value of the game. Formally, consider a *value function* $v : 2^{[n]} \to \mathbb{R}$ defined on sets $S \subseteq [n]$, where $[n]$ denotes $\{1, \ldots, n\}$. The probabilistic value for player $i \in [n]$ is

$$\phi_i(v) = \sum_{S \subseteq [n] \setminus \{i\}} p_{|S|}[v(S \cup \{i\}) - v(S)] \tag{1}$$

where $\mathbf{p} = [p_0, \ldots, p_{n-1}] \in [0, 1]^n$ is a set of probabilistic weights that satisfy $\sum_{\ell=0}^{n-1} \binom{n-1}{\ell} p_\ell = 1$. We can interpret the $i$th probabilistic value as the *average* marginal contribution of player $i$ to random set $S$, where the distribution over set sizes is specified by $\mathbf{p}$. Different choices of $\mathbf{p}$ yield different variants of probabilistic values [KZ22a, KZ22b, LY24c]. For example, to obtain the ubiquitous Shapley values, set $p_\ell = \frac{1}{n} \binom{n-1}{\ell}^{-1}$, and to obtain Banzhaf values, set $p_\ell = 1/2^{n-1}$ for all $\ell$.

39th Conference on Neural Information Processing Systems (NeurIPS 2025).

Our paper addresses the problem of computing $\phi_1, \ldots, \phi_n$ in full generality for any $\mathbf{p}$. The topic of which weights are best for a given application has received significant attention. Some prior work focuses on axiomatic approaches for choosing $\mathbf{p}$. For example, all probabilistic values satisfy three desirable properties: *null player*, *symmetry*, and *linearity* (see [Web88] for a detailed discussion). Shapley values satisfy an additional *efficiency* property [Sha51] and Banzhaf values satisfy a *2-efficiency* property that might be desirable when there are non-linear interactions between players [Pen46, BI64]. Generalizations of these values include Beta Shapley [KZ22a] values and weighted Banzhaf values [LY24c]. See Appendix B for more on these generalizations.

Regardless of how $\mathbf{p}$ is chosen, the meaning of the probabilistic values depends on how the value function, $v$, is defined. For example, a common task in explainable AI is to attribute a model prediction (for a given input) to features [LL17]. Here, $v(S)$, is the prediction made when using just the subset of features corresponding to $S$.[1] Probabilistic values are also used in data attribution tasks, where $v(S)$ corresponds to the model loss when training with a given subset of data [GZ19, WMSJ25]. In these applications and others, evaluating $v$ is expensive, as it requires re-running or possibly even re-training a model. As in prior work on efficient probabilistic value estimation, we thus focus on algorithms that estimate $\phi_1, \ldots, \phi_n$ using as few evaluations of $v$ as possible. We view these evaluations as black-box, designing algorithms that are agnostic to the particular value function $v$, and can thus be applied in a wide range of downstream applications.

## 1.1 Efficiently Computing Probabilistic Values

For general value functions, exactly computing probabilistic values requires exponential time, as the summation in Equation (1) involves $O(2^n)$ terms. When $v$ is a highly structured function, like a linear function or decision tree, more efficient algorithms exist [LL17, LEL18, LEC$^+$20, KMM$^+$22]. However, given the complexity of modern machine learning models, most prior work focuses on approximation algorithms.

The standard method is to approximate the summation in Equation (1) via a Monte Carlo estimate obtained from a weighted sample of sets that do not contain $i$ [KZ22a, KZ22b, LY24b]. Concretely, assume for simplicity that we sample a collection of subsets, $\mathcal{S}_i$, by drawing samples with replacement from a distribution with density $\mathcal{D} : 2^{[n] \setminus \{i\}} \to [0, 1]$.[2] We then compute the unbiased estimate:

$$\tilde{\phi}_i^{\mathrm{MC}} = \frac{1}{|\mathcal{S}_i|} \sum_{S \in \mathcal{S}_i} [v(S \cup \{i\}) - v(S)] \frac{p_{|S|}}{\mathcal{D}(S)} \tag{2}$$

We have that $\mathbb{E}[\tilde{\phi}_i^{\mathrm{MC}}] = \phi_i$, and the estimator's variance depends on the choice of sampling distribution $\mathcal{D}$, as well as $[v(S \cup \{i\}) - v(S)]^2$ for all $S \subseteq [n]$. In addition to high-variance in practice[3], a downside of Monte Carlo estimators is that it is difficult to "reuse" samples between indices $1, \ldots, n$, as each term in Equation (2) requires evaluating both $v(S \cup \{i\})$ and $v(S)$ for a particular $i$. Several methods address this issue via "sample reuse" [CGT09]. One technique especially relevant to our work is the *maximum sample reuse* (MSR) method, which was originally applied to Banzhaf values [WJ23], but generalizes naturally to all probabilistic values [KBMH24, LY24a, LY24b]. The MSR method draws a single collection of subsets, $\mathcal{S}$, according to $\mathcal{D} : 2^{[n]} \to [0, 1]$, and computes the estimate:

$$\tilde{\phi}_i^{\mathrm{MSR}} = \frac{1}{|\mathcal{S}|} \sum_{S \in \mathcal{S}} v(S) \frac{p_{|S|-1} \mathbb{1}[i \in S] - p_{|S|} \mathbb{1}[i \notin S]}{\mathcal{D}(S)}. \tag{3}$$

It can be checked that we still have $\mathbb{E}[\tilde{\phi}_i^{\mathrm{MSR}}] = \phi_i$ for all $i$. Moreover, every evaluation of the value function, $v(S)$, contributes to the estimate for *all* $i \in [n]$, so we achieve maximum sample reuse. However, the variance of MSR methods scales as a weighted sum of $[v(S)]^2$, which is generally much larger than the difference between nearby values $[v(S \cup \{i\}) - v(S)]^2$.

**Beyond Monte Carlo.** Given the high variance of Monte Carlo methods, an alternate approach based on *regression* has become popular for the special case of Shapley values. In particular, Shapley values

---

[1]Since most models in machine learning require a full set of input features, features not in $S$ are replaced with either a mean value or random value from the training dataset as a baseline [JMB20, LEL18].

[2]In order to efficiently sample, $\mathcal{D}$ typically assigns the same density to subsets of the same size.

[3]In general, variance scales with $1/\sqrt{|\mathcal{S}_i|}$, i.e., only as the inverse root of the number of samples.

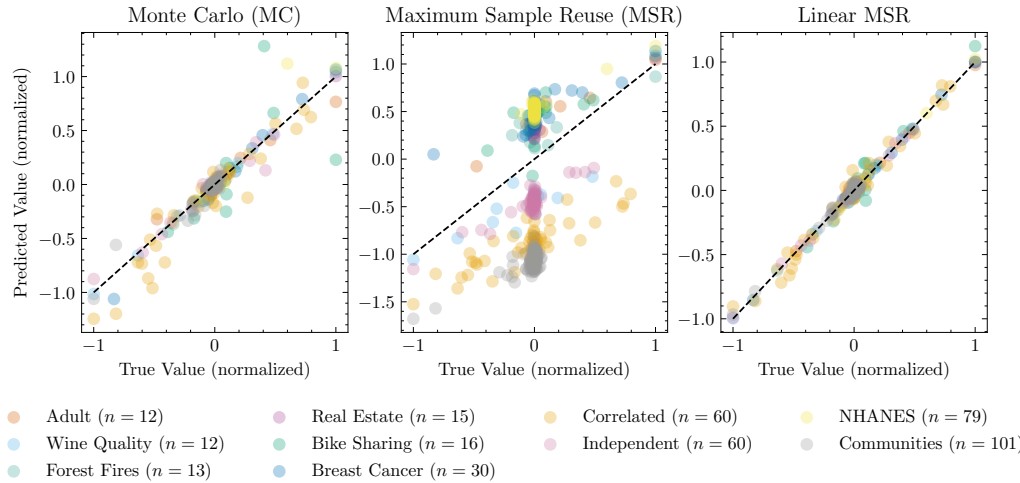

Figure 1: Predicted versus true (normalized) Shapley values for three unbiased estimators given a fixed number of black-box evaluations of the value function, $v$. Each point represents one feature's estimated vs true Shapley value on one dataset. The Monte Carlo estimator uses each sample to estimate only one Shapley value, but has variance that depends on the difference in values between neighboring sets, i.e., $[v(S \cup \{i\}) - v(S)]^2$. The Maximum Sample Reuse (MSR) estimator reuses samples, but has larger variance that depends on the magnitude of the values, i.e., $[v(S)]^2$. Our Regression MSR estimators reuse samples *and* have smaller variance that depends on how well a learned function $f$ fits the value function $v$, i.e., $[v(S) - f(S)]^2$. Even taking $f$ to be linear gives excellent performance (we call this method Linear MSR). Taking $f$ to be a decision-tree model (Tree MSR) can produce even better estimates for large sample sizes, as shown in Figure 2.

are the unique solution to a particular overdetermined linear regression problem [CGKR88]:

$$\boldsymbol{\phi} = [\phi_1, \ldots, \phi_n] = \underset{\mathbf{x}:\langle \mathbf{x}, \mathbf{1} \rangle = v([n]) - v(\emptyset)}{\arg \min} \|\mathbf{A}\mathbf{x} - \mathbf{b}\|_W, \tag{4}$$

where $\mathbf{A} \in \mathbb{R}^{2^n \times n}$ is a specific structured matrix whose rows correspond to sets $S \subseteq [n]$, $\mathbf{b} \in \mathbb{R}^{2^n}$ is vector whose entries equal $v(S) - v(\emptyset)$, and $\| \cdot \|_W$ is a weighted $\ell_2$ norm.

The ubiquitous Kernel SHAP algorithm [LL17, CL21] takes advantage of the regression formulation by *approximately solving* Equation (4) using a subsample of constraints (and corresponding entries in $\mathbf{b}$), each of which requires evaluating $v(S)$ for a single subset $S$. This approach was recently improved by incorporating leverage score sampling [Sar06, SS11], resulting in the state-of-the-art Leverage SHAP method [MW25]. In addition to inherent sample reuse, the empirical effectiveness of Kernel SHAP and Leverage SHAP seems related to the fact that the accuracy of both methods depends on how well $v$ is approximated by a linear function. Indeed, it can be shown that if $v$ is exactly linear, both methods return exact Shapley values after just $n$ function evaluations [MW25]. However, even when $v$ is not linear, there are theoretical guarantees on the performance of Kernel SHAP and Leverage SHAP [MW25, CSV+25].

The Kernel SHAP approach has been extended to Banzhaf values [LWK+25], thanks to a similarly elegant regression formulation [HH92]. However, extensions to more general probabilistic values have been less effective, failing to outperform Monte Carlo methods [LZL+22, LY24a, LY24b]. A key challenge is that, due to the lack of an efficiency property, generalized linear regression formulations for probabilistic values typically require estimating the *sum* of these values, which introduces another source of error [RVZ98]. Moreover, even for Shapley and Banzhaf values, a drawback of regression-based methods is that they fail to provide an unbiased estimate for each $\phi_i$. Attempts to fix this issue have generally led to estimates with much higher variance [CL21].

### 1.2   Our Contributions

We introduce a method called *Regression MSR* for leveraging regression to approximate probabilistic values. In contrast to previous work on regression methods, Regression MSR leads to estimates that

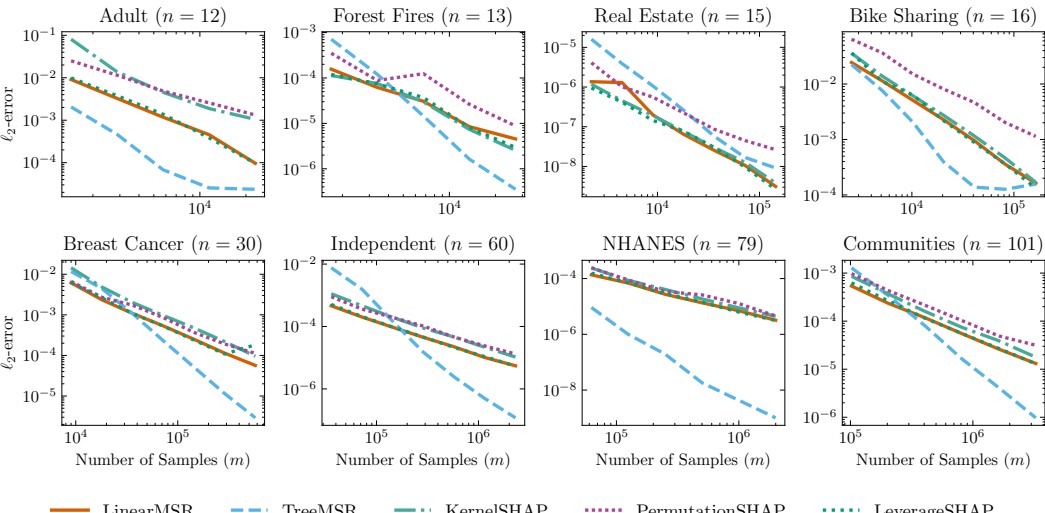

Figure 2: Average $\ell_2$-error between estimated and true Shapley values as a function of sample size $m$ (number of evaluations of $v$) for various datasets. The lines report the mean error over 100 runs, and $m = 10n, 20n, 40n, 80n, 160n, 320n, 640n$. Linear MSR consistently performs comparably to the prior state-of-the-art Leverage SHAP. Meanwhile, the performance of Tree MSR depends on how well the tree-based model approximates the value function; with more samples, it can even outperform Leverage SHAP by several orders of magnitude.

are unbiased and easily extend to all probabilistic values. Moreover, the method can take advantage of non-linear regression methods like XGBoost [CG16] and other decision-tree models.

Instead of starting with a custom linear regression formulation for a given type of probabilistic value, Regression MSR uses regression as a *variance reduction* method for Monte Carlo approximation, and specifically, for the Maximum Sample Reuse method introduced earlier. Concretely, learning from a small number of random subsets, we start by approximating the value function $v$ using a simpler function, $f$. Using the fact that the probabilistic values are linear — i.e., $\phi_i(v) = \phi_i(f) + \phi_i(v - f)$ for any $f$ — we propose to return the estimator:

$$\tilde{\phi}_i = \phi_i(f) + \frac{1}{|\mathcal{S}|} \sum_{S \in \mathcal{S}} [v(S) - f(S)] \frac{p_{|S|-1} \mathbb{1}[i \in S] - p_{|S|} \mathbb{1}[i \notin S]}{\mathcal{D}(S)}. \tag{5}$$

Since the MSR estimator (second term) is consistent — i.e., returns the true Shapley values when run on all subsets — the Regression MSR estimator is too. Further, it can be checked that $\mathbb{E}\left[\tilde{\phi}_i\right] = \phi_i$ for any fixed $f$ (e.g., one learned using samples not in $\mathcal{S}$). That is, the method is unbiased. Moreover, like the biased Kernel and Leverage SHAP methods, the variance of $\tilde{\phi}_i$ depends on $[v(S) - f(S)]^2$ (see Section 2 for details). So, our method is more accurate than the standard MSR method when we can obtain a good approximation to $v$.

The benefit of our approach is clear in Figure 1, where we take $f$ to be a linear approximation to $v$ and use Equation (5) to estimate Shapley values. However, there is also the potential to go beyond linear approximations. Observe that, to evaluate $\tilde{\phi}_i$, the function $f$ does not need to be a linear. Indeed, we can use *any approximation for which the $\phi_i(f)$ term in Equation (5) can be computed efficiently*. That is, any function family that admits efficient probabilistic value computation. Importantly, this includes a wide variety of functions based on decision trees. Concretely, in Appendix C, we show how to efficiently compute probabilistic values for any linear mixture of decision trees.[4]

---

[4]Efficient methods for computing Shapley and Banzhaf values for decision trees were previously known [LEL18, LEC+20, KMM+22]. However, they were based on a particular summation property that does not hold for general probabilistic values; if a value function only has contributions from $n' < n$ players, the

We leverage this observation to learn tree-based approximations to $v$ using powerful models like XGBoost (for the purposes of our experiments, we call this variant of our algorithm Tree MSR). For the well-studied Shapley values, we find that Tree MSR achieves state-of-the-art performance, especially when there are enough samples for the tree-based model to learn an accurate fit, see e.g., Figure 2. In particular, Tree MSR can yield estimates with average error that is $2.6\times$ lower than the prior state-of-the-art Leverage SHAP estimator (see Table 1). For general probabilistic values, Tree MSR gives up to $215\times$ lower average error than the best estimator from prior work (see Figure 6 and Table 3).

Concurrent to our work, [BAK$^+$25] introduce Proxy SPEX for estimating probabilistic values. Like Tree MSR, they fit gradient boosted trees to the value function $v$. However, instead of computing the probabilistic values of the trees, they extract the most influential Fourier terms and compute the probabilistic values of the Fourier representation. In terms of performance, Proxy SPEX outperforms Kernel SHAP for the low sample regime with budget $m \leq 5n$, but Kernel SHAP is more accurate for moderate and larger sample regimes [BAK$^+$25]. Tree MSR underperforms Kernel SHAP (and hence Proxy SPEX) in the low sample regime, but generally outperforms Leverage SHAP in larger sample regimes (see e.g., Figure 2). Proxy SPEX is neither consistent nor unbiased, unlike Regression MSR.

## 2  Regression MSR

In this section, we present our Regression MSR method, which combines the benefits of Monte Carlo and regression-based estimators. In particular, Regression MSR produces estimates that are unbiased (like Monte Carlo methods), reuses every sample for each estimate (like Maximum Sample Reuse and regression-based methods), and achieves lower variance when a learned approximation is accurate (like regression-based methods). Unlike prior linear regression-based methods, Regression MSR successfully extends to any probabilistic value, and can harness the accuracy of richer function classes like regression trees.

The pseudocode of Regression MSR appears in Algorithm 1. We separate the samples used to train from the samples in the final prediction; this both ensures the estimator is unbiased, and allows us to give strong theoretical guarantees in Theorem 2.1. First, the algorithm partitions $m$ samples into $k$ collections of samples $\mathcal{S}^{(1)}, \ldots, \mathcal{S}^{(k)}$. The algorithm then proceeds in three phases, repeated for each $\mathcal{S}^{(\ell)}$: During the first phase, Regression MSR learns an approximation $f^{(\ell)}$ to the value function $v$, on all samples that are *not* in $\mathcal{S}^{(\ell)}$. In the second phase, the probabilistic values $\phi_i(f^{(\ell)})$ are computed for all $i$. (We run Regression MSR with linear or tree-based methods so that computing their probabilistic values is efficient.) Finally, the algorithm uses the learned function to reduce the variance of the MSR estimates on the samples in $\mathcal{S}^{(\ell)}$.

Theorem 2.1 gives theoretical guarantees on the performance of Regression MSR. For a constant error constraint $\epsilon > 0$ and failure probability $\delta > 0$, Regression MSR uses a linear number of samples to produce estimates with $\ell_2$-norm error that depends on a natural weighted squared error between the value function and our worst learned function. We present the guarantee for any sampling distribution $\mathcal{D}$ over subsets, and, below, discuss our suggested choice of this distribution.

**Theorem 2.1** (Regression-Adjustment Guarantee)**.** *The estimates produced by Algorithm 1 are unbiased estimates of the probabilistic values. Further, let $\epsilon, \delta > 0$, and $f_{\max}$ be the learned function $f^{(\ell)}$ with largest generalization error over $\ell \in [k]$. When run with $m = O(n\frac{1}{\epsilon\delta})$ samples, Algorithm 1 produces estimates that satisfy, with probability $1 - \delta$,*

$$\|\tilde{\phi} - \phi\|_2^2 \leq \epsilon \sum_{S \subseteq [n]} [v(S) - f_{\max}(S)]^2 \frac{p_{|S|}^2(1 - \frac{|S|}{n}) + p_{|S|-1}^2 \frac{|S|}{n}}{\mathcal{D}(S)}. \tag{6}$$

Algorithm 1 can be used to make the estimates of any regression-based estimator unbiased while preserving its variance. To this end, we purposefully do not specify the sampling distribution $\mathcal{D}$ or the function class $f$. We next discuss two choices for how to select the model, $f$, and collect samples.

---

Shapley/Banzhaf value on the induced game of those $n'$ players is the same as the Shapley/Banzhaf value on the original value function with all $n$ players. Our approach in Appendix C is based on an alternative way of viewing tree-based models that avoids the need for this property.

---
**Algorithm 1** Regression Maximum Sample Reuse
---

1: **Input:** number of players $n$, number of samples $m$, value function $v : 2^{[n]} \to \mathbb{R}$, probabilistic weights $\mathbf{p} \in [0,1]^n$, probability density function for sampling $\mathcal{D} : 2^{[n]} \to [0,1]$, number of splits $k$
2: **Output:** Estimated probabilistic values $\tilde{\phi}_1, \ldots, \tilde{\phi}_n$
3: Sample $\mathcal{S}$, consisting of $m$ subsets drawn with (or without) replacement from $\mathcal{D}$.
4: Randomly partition $\mathcal{S}$ into $\mathcal{S}^{(1)}, \ldots, \mathcal{S}^{(k)}$.
5: **for** $\ell \in \{1, \ldots, k\}$ **do**
6:     For $i \in [n]$, initialize $\tilde{\phi}_i^{(\ell)} \leftarrow 0$.
7:     Learn $f^{(\ell)} : 2^{[n]} \to \mathbb{R}$ to minimize loss

$$\sum_{S \in \cup_{\ell' \neq \ell} \mathcal{S}^{(\ell')}} [v(S) - f(S)]^2.$$

8:     For all $i \in [n]$, compute probabilistic values $\phi_i(f^{(\ell)})$. ▷ Efficient for linear/tree-based models.

9:     For all $i \in [n]$, compute

$$\tilde{\phi}_i^{(\ell)} \leftarrow \phi_i(f^{(\ell)}) + \frac{1}{|\mathcal{S}^{(\ell)}|} \sum_{S \in \mathcal{S}^{(\ell)}} [v(S) - f^{(\ell)}(S)] \frac{p_{|S|-1} \mathbb{1}[i \in S] - p_{|S|} \mathbb{1}[i \notin S]}{\mathcal{D}(S)}.$$

10: **end for**
11: For all $i \in [n]$, compute final estimate $\tilde{\phi}_i \leftarrow \frac{1}{k} \sum_\ell \tilde{\phi}_i^{(\ell)}$.
12: **return** $\tilde{\phi}_1, \ldots, \tilde{\phi}_n$

---

**Linear MSR** The simplest choice for $f$ is a linear model. As discussed in the introduction, there is extensive prior work on special linear regression formulations for Shapley values [CGKR88, LL17, CL21, MW25]. Using a linear function for variance reduction in MSR offers a natural alternative to these methods, and adds negligible computational overhead, yet tends to show superior performance on most datasets (see, e.g., Table 1). When applying the method to Shapley values specifically, we use the existing state-of-the art Leverage SHAP method (which samples via leverage scores) to fit the learned function. We similarly use the linear regression-based Kernel Banzhaf algorithm [LWK+25] when estimating Banzhaf values. For general probabilistic values, Linear MSR fits a linear model with the sampling distribution described below, and uses its predictions to adjust the final estimates.

**Tree MSR** Beyond linear models, tree-based regression models like XGBoost can learn more accurate approximations. As discussed in the introduction, it is known how to efficiently compute the Shapley values [LEC+20] and Banzhaf values [KMM+22] of trees; however, it was previously unclear how to generalize these approaches to probabilistic values: the Shapley/Banzhaf methods use the property that the probabilistic value of a value function on $n$ players is the probabilistic value of the extended value function on $n' > n$ players, as long as the additional $n' - n$ players always contribute nothing. This property unfortunately does not hold for all probabilistic values; e.g., consider probabilistic weights $\mathbf{p}$ that are independently sampled (and normalized) for each number of players $n$ and $n'$. Instead, efficiently computing the probabilistic values of trees requires a subtly different approach, which we describe in Appendix C. With this approach in hand, we can fit $v$ with a tree-based model like XGBoost and efficiently compute its probabilistic values for the final estimate.

**Sampling Distribution** When Algorithm 1 is not run on top of another regression-based estimator, we choose the sampling distribution so that the function $f$ is directly trained to minimize the error bound in Theorem 2.1. That is, we sample each set with probability proportional to

$$\sqrt{p_{|S|}^2 (1 - \frac{|S|}{n}) + p_{|S|-1}^2 \frac{|S|}{n}}.$$

Then the error bound in Theorem 2.1 is proportional to

$$\sum_{S \subseteq [n]} [v(S) - f(S)]^2 \sqrt{p_{|S|}^2 (1 - \frac{|S|}{n}) + p_{|S|-1}^2 \frac{|S|}{n}}$$

Table 1: Summary statistics of the average $\ell_2$-norm error between estimated and true Shapley values for all datasets listed in Appendix G. All estimators are run with $m = 40n$ samples. Tree MSR achieves average error that is $6.5\times$ lower than Permutation SHAP, $3.8\times$ lower than Kernel SHAP, and $2.6\times$ lower than the prior state-of-the-art Leverage SHAP. We emphasize that Tree MSR gives even better performance for larger sample sizes, as shown in Figure 2. We follow Olympic medal convention: gold , silver and bronze signify first, second and third best performance, respectively.

| | Adult | Forest Fires | Real Estate | Bike Sharing | Breast Cancer | Independent | NHANES | Communities | Mean |
|---|---|---|---|---|---|---|---|---|---|
| **LinearMSR** | | | | | | | | | |
| Mean | $1.18 \times 10^{-3}$ | $3.07 \times 10^{-5}$ | $2.00 \times 10^{-7}$ | $5.07 \times 10^{-3}$ | $1.09 \times 10^{-3}$ | $9.49 \times 10^{-5}$ | $2.73 \times 10^{-5}$ | $1.17 \times 10^{-4}$ | $9.51 \times 10^{-4}$ |
| 1st Quartile | $2.99 \times 10^{-4}$ | $4.72 \times 10^{-7}$ | $2.46 \times 10^{-8}$ | $9.72 \times 10^{-4}$ | $2.14 \times 10^{-4}$ | $5.38 \times 10^{-5}$ | $2.18 \times 10^{-10}$ | $4.76 \times 10^{-5}$ | $1.98 \times 10^{-4}$ |
| 2nd Quartile | $7.67 \times 10^{-4}$ | $2.75 \times 10^{-6}$ | $5.91 \times 10^{-8}$ | $2.85 \times 10^{-3}$ | $1.02 \times 10^{-3}$ | $6.64 \times 10^{-5}$ | $2.49 \times 10^{-6}$ | $9.46 \times 10^{-5}$ | $6.00 \times 10^{-4}$ |
| 3rd Quartile | $1.52 \times 10^{-3}$ | $6.00 \times 10^{-6}$ | $1.82 \times 10^{-7}$ | $6.37 \times 10^{-3}$ | $1.71 \times 10^{-3}$ | $1.06 \times 10^{-4}$ | $2.88 \times 10^{-5}$ | $1.45 \times 10^{-4}$ | $1.24 \times 10^{-3}$ |
| **TreeMSR** | | | | | | | | | |
| Mean | $6.77 \times 10^{-5}$ | $1.45 \times 10^{-5}$ | $1.07 \times 10^{-6}$ | $2.04 \times 10^{-3}$ | $1.08 \times 10^{-3}$ | $1.47 \times 10^{-4}$ | $1.95 \times 10^{-7}$ | $7.93 \times 10^{-5}$ | $4.29 \times 10^{-4}$ |
| 1st Quartile | $1.79 \times 10^{-5}$ | $1.32 \times 10^{-6}$ | $9.50 \times 10^{-8}$ | $6.23 \times 10^{-4}$ | $2.37 \times 10^{-4}$ | $2.40 \times 10^{-5}$ | $2.99 \times 10^{-10}$ | $1.78 \times 10^{-5}$ | $1.15 \times 10^{-4}$ |
| 2nd Quartile | $4.12 \times 10^{-5}$ | $3.55 \times 10^{-6}$ | $1.97 \times 10^{-7}$ | $1.28 \times 10^{-3}$ | $5.51 \times 10^{-4}$ | $8.20 \times 10^{-5}$ | $8.89 \times 10^{-10}$ | $3.58 \times 10^{-5}$ | $2.50 \times 10^{-4}$ |
| 3rd Quartile | $9.03 \times 10^{-5}$ | $1.01 \times 10^{-5}$ | $1.40 \times 10^{-6}$ | $2.44 \times 10^{-3}$ | $1.23 \times 10^{-3}$ | $1.70 \times 10^{-4}$ | $3.91 \times 10^{-9}$ | $5.70 \times 10^{-5}$ | $5.00 \times 10^{-4}$ |
| **KernelSHAP** | | | | | | | | | |
| Mean | $4.55 \times 10^{-3}$ | $2.98 \times 10^{-5}$ | $1.93 \times 10^{-7}$ | $6.12 \times 10^{-3}$ | $2.01 \times 10^{-3}$ | $1.97 \times 10^{-4}$ | $4.08 \times 10^{-5}$ | $1.59 \times 10^{-4}$ | $1.64 \times 10^{-3}$ |
| 1st Quartile | $5.14 \times 10^{-4}$ | $3.08 \times 10^{-7}$ | $1.04 \times 10^{-9}$ | $1.40 \times 10^{-3}$ | $6.87 \times 10^{-4}$ | $1.09 \times 10^{-4}$ | $1.60 \times 10^{-16}$ | $7.03 \times 10^{-5}$ | $3.47 \times 10^{-4}$ |
| 2nd Quartile | $8.59 \times 10^{-4}$ | $3.05 \times 10^{-6}$ | $3.50 \times 10^{-8}$ | $4.00 \times 10^{-3}$ | $1.89 \times 10^{-3}$ | $1.64 \times 10^{-4}$ | $3.10 \times 10^{-6}$ | $1.27 \times 10^{-4}$ | $8.81 \times 10^{-4}$ |
| 3rd Quartile | $2.84 \times 10^{-3}$ | $7.30 \times 10^{-6}$ | $1.59 \times 10^{-7}$ | $7.91 \times 10^{-3}$ | $2.98 \times 10^{-3}$ | $2.80 \times 10^{-4}$ | $3.97 \times 10^{-5}$ | $2.25 \times 10^{-4}$ | $1.79 \times 10^{-3}$ |
| **PermutationSHAP** | | | | | | | | | |
| Mean | $4.86 \times 10^{-3}$ | $1.25 \times 10^{-4}$ | $5.58 \times 10^{-7}$ | $1.51 \times 10^{-2}$ | $1.73 \times 10^{-3}$ | $1.96 \times 10^{-4}$ | $3.43 \times 10^{-5}$ | $2.14 \times 10^{-4}$ | $2.78 \times 10^{-3}$ |
| 1st Quartile | $1.65 \times 10^{-3}$ | $8.54 \times 10^{-7}$ | $3.64 \times 10^{-9}$ | $3.13 \times 10^{-3}$ | $2.97 \times 10^{-4}$ | $6.96 \times 10^{-5}$ | $1.60 \times 10^{-16}$ | $5.87 \times 10^{-5}$ | $6.50 \times 10^{-4}$ |
| 2nd Quartile | $3.84 \times 10^{-3}$ | $4.83 \times 10^{-6}$ | $4.90 \times 10^{-8}$ | $5.97 \times 10^{-3}$ | $1.05 \times 10^{-3}$ | $1.70 \times 10^{-4}$ | $2.10 \times 10^{-6}$ | $1.61 \times 10^{-4}$ | $1.40 \times 10^{-3}$ |
| 3rd Quartile | $7.68 \times 10^{-3}$ | $1.52 \times 10^{-5}$ | $2.69 \times 10^{-7}$ | $1.92 \times 10^{-2}$ | $1.97 \times 10^{-3}$ | $2.77 \times 10^{-4}$ | $2.09 \times 10^{-5}$ | $2.78 \times 10^{-4}$ | $3.68 \times 10^{-3}$ |
| **LeverageSHAP** | | | | | | | | | |
| Mean | $1.38 \times 10^{-3}$ | $3.71 \times 10^{-5}$ | $1.44 \times 10^{-7}$ | $6.32 \times 10^{-3}$ | $1.08 \times 10^{-3}$ | $9.62 \times 10^{-5}$ | $2.83 \times 10^{-5}$ | $1.15 \times 10^{-4}$ | $1.13 \times 10^{-3}$ |
| 1st Quartile | $3.35 \times 10^{-4}$ | $3.07 \times 10^{-7}$ | $7.88 \times 10^{-10}$ | $1.05 \times 10^{-3}$ | $2.74 \times 10^{-4}$ | $5.32 \times 10^{-5}$ | $1.60 \times 10^{-16}$ | $4.41 \times 10^{-5}$ | $2.20 \times 10^{-4}$ |
| 2nd Quartile | $6.62 \times 10^{-4}$ | $2.22 \times 10^{-6}$ | $3.21 \times 10^{-8}$ | $2.73 \times 10^{-3}$ | $1.09 \times 10^{-3}$ | $7.30 \times 10^{-5}$ | $2.70 \times 10^{-6}$ | $9.36 \times 10^{-5}$ | $5.81 \times 10^{-4}$ |
| 3rd Quartile | $1.62 \times 10^{-3}$ | $5.13 \times 10^{-6}$ | $1.29 \times 10^{-7}$ | $7.03 \times 10^{-3}$ | $1.46 \times 10^{-3}$ | $1.08 \times 10^{-4}$ | $2.62 \times 10^{-5}$ | $1.52 \times 10^{-4}$ | $1.30 \times 10^{-3}$ |

which, by design, is the *expected* loss used to train $f$.

**Bias vs. Accuracy vs. Runtime** Regression MSR produces unbiased estimates by training $k$ functions, each on a $(k-1)/k$ fraction of the available samples and evaluating on the held-out $1/k$. This creates a trade-off: increasing $k$ improves accuracy (each function has a larger training set) but raises computational cost. In our experiments, we set $k = 10$, meaning that each function is trained on $90\%$ of the data. Thanks to efficient solvers (e.g., least squares or XGBoost), this setup maintains fast runtimes while delivering high accuracy.

**Practical Simplification** Unless a small bias term (similar to that of Leverage SHAP or Kernel SHAP) is unacceptable, we recommend simplifying the Regression MSR algorithm: Train a single function—and build the final estimate with it—on all samples. While this introduces a small bias (and breaks the theoretical guarantees), the resulting algorithm runs faster and is generally more accurate than the version described in this paper.

## 3 Experiments

In this section, we describe our experiments on eight datasets. Overall, we find that Linear MSR and Tree MSR give state-of-the-art performance for almost all datasets and sample budgets.[5]

**Value Function** For evaluation, we focus on the explainable AI feature attribution task, but emphasize that our methods can be applied to any application involving probabilistic values, as we only require black-box access to the value function $v$. Concretely, we train a model on a dataset, and attribute the prediction the model makes on a given *explicand* point $\mathbf{x}^e \in \mathbb{R}^n$ to its $n$ input features. We consider the *interventional* definition of $v$, where the explanation is relative to a *baseline* point $\mathbf{x}^b \in \mathbb{R}^n$: For a set $S$, let $\mathbf{x}^S$ be the point where the $i$th feature is $x_i^e$ if $i \in S$ and $x_i^b$ otherwise. Then, the value function $v(S)$ is the model's prediction on $\mathbf{x}^S$. There is also a conditional version of feature attribution, where the features not in $S$ are drawn from a background dataset [LL17, LEL18]. However, we choose to focus on the interventional version since it is more efficient to compute $v(S)$, and the resulting probabilistic values are more interpretable [JMB20].

**Ground Truth Probabilistic Values** For small datasets with $n < 30$, we use a neural network model and compute the true probabilistic values through enumeration. For larger datasets with $n \geq 30$ where

---

[5]The code is available at `https://github.com/rtealwitter/regressionMSR`.

Probabilistic Values: Error vs Sample Complexity (Regression Forest Ground Truth)

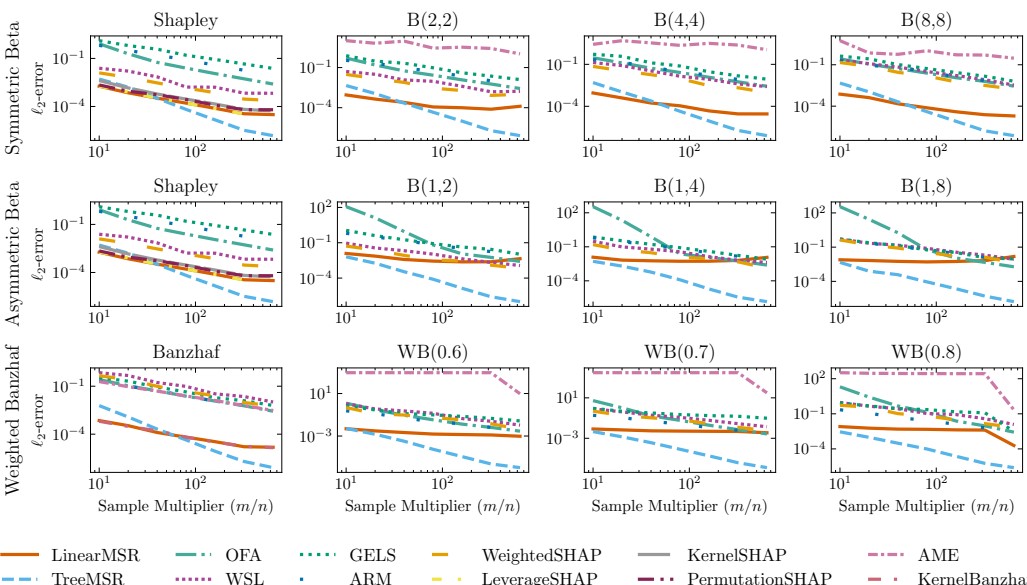

Figure 3: Average error between the estimated and true probabilistic values as a function of sample size. Each subplot shows results for a different probabilistic value with the error averaged over all large datasets ($n \geq 30$), for which we used the tree-based method described in Appendix C. The lines report the mean error over 10 runs. Tree MSR gives the best performance, often by several orders of magnitude when $m$ is large.

Table 2: Summary statistics of the $\ell_2$-norm error between estimated and true probabilistic values when $m = 40n$. We summarize the error over large datasets ($n \geq 30$), for which we use the tree-based method described in Appendix C to compute the true probabilistic values. On average over all probabilistic values, Tree MSR produces estimates with mean error that is $215\times$ lower than the best estimator from prior work.

| | B(1,1) | B(2,2) | B(4,4) | B(8,8) | B(1,2) | B(1,4) | B(1,8) | WB(0.5) | WB(0.6) | WB(0.7) | WB(0.8) | WB(0.9) | Mean |
|---|---|---|---|---|---|---|---|---|---|---|---|---|---|
| **LinearMSR** | | | | | | | | | | | | | |
| Mean | $3.03 \times 10^{-4}$ | $2.50 \times 10^{-4}$ | $1.72 \times 10^{-4}$ | $1.48 \times 10^{-4}$ | $3.80 \times 10^{-3}$ | $5.85 \times 10^{-3}$ | $5.66 \times 10^{-3}$ | $1.45 \times 10^{-4}$ | $2.13 \times 10^{-3}$ | $5.46 \times 10^{-3}$ | $4.91 \times 10^{-3}$ | $1.13 \times 10^{-3}$ | $2.50 \times 10^{-3}$ |
| 1st Quartile | $6.26 \times 10^{-6}$ | $8.70 \times 10^{-6}$ | $5.99 \times 10^{-6}$ | $6.65 \times 10^{-6}$ | $5.59 \times 10^{-4}$ | $5.25 \times 10^{-4}$ | $3.25 \times 10^{-4}$ | $5.29 \times 10^{-6}$ | $7.66 \times 10^{-4}$ | $1.53 \times 10^{-3}$ | $2.51 \times 10^{-4}$ | $1.02 \times 10^{-4}$ | $3.41 \times 10^{-4}$ |
| 2nd Quartile | $5.90 \times 10^{-5}$ | $4.74 \times 10^{-5}$ | $4.42 \times 10^{-5}$ | $3.16 \times 10^{-5}$ | $1.06 \times 10^{-3}$ | $1.03 \times 10^{-3}$ | $7.56 \times 10^{-4}$ | $3.81 \times 10^{-5}$ | $1.54 \times 10^{-3}$ | $4.27 \times 10^{-3}$ | $7.39 \times 10^{-4}$ | $2.79 \times 10^{-4}$ | $8.25 \times 10^{-4}$ |
| 3rd Quartile | $1.46 \times 10^{-4}$ | $1.23 \times 10^{-4}$ | $1.37 \times 10^{-4}$ | $8.92 \times 10^{-5}$ | $3.01 \times 10^{-3}$ | $3.11 \times 10^{-3}$ | $2.47 \times 10^{-3}$ | $8.76 \times 10^{-5}$ | $2.94 \times 10^{-3}$ | $7.17 \times 10^{-3}$ | $5.21 \times 10^{-3}$ | $8.08 \times 10^{-4}$ | $2.11 \times 10^{-3}$ |
| **TreeMSR** | | | | | | | | | | | | | |
| Mean | $2.64 \times 10^{-4}$ | $2.21 \times 10^{-4}$ | $2.41 \times 10^{-4}$ | $2.06 \times 10^{-4}$ | $3.47 \times 10^{-4}$ | $5.91 \times 10^{-4}$ | $3.83 \times 10^{-4}$ | $2.09 \times 10^{-4}$ | $3.25 \times 10^{-4}$ | $3.67 \times 10^{-4}$ | $3.06 \times 10^{-4}$ | $2.18 \times 10^{-4}$ | $3.07 \times 10^{-4}$ |
| 1st Quartile | $2.48 \times 10^{-6}$ | $1.29 \times 10^{-6}$ | $9.50 \times 10^{-7}$ | $1.23 \times 10^{-6}$ | $1.77 \times 10^{-6}$ | $4.10 \times 10^{-6}$ | $1.08 \times 10^{-5}$ | $1.17 \times 10^{-6}$ | $8.75 \times 10^{-7}$ | $7.87 \times 10^{-7}$ | $5.09 \times 10^{-6}$ | $7.61 \times 10^{-6}$ | $3.18 \times 10^{-6}$ |
| 2nd Quartile | $4.77 \times 10^{-5}$ | $3.61 \times 10^{-5}$ | $2.91 \times 10^{-5}$ | $2.94 \times 10^{-5}$ | $5.24 \times 10^{-5}$ | $8.58 \times 10^{-5}$ | $8.84 \times 10^{-5}$ | $2.97 \times 10^{-5}$ | $3.14 \times 10^{-5}$ | $3.11 \times 10^{-5}$ | $3.82 \times 10^{-5}$ | $3.59 \times 10^{-5}$ | $4.46 \times 10^{-5}$ |
| 3rd Quartile | $3.28 \times 10^{-4}$ | $2.31 \times 10^{-4}$ | $2.12 \times 10^{-4}$ | $1.72 \times 10^{-4}$ | $3.37 \times 10^{-4}$ | $5.80 \times 10^{-4}$ | $3.54 \times 10^{-4}$ | $1.85 \times 10^{-4}$ | $2.36 \times 10^{-4}$ | $2.85 \times 10^{-4}$ | $3.23 \times 10^{-4}$ | $1.97 \times 10^{-4}$ | $2.87 \times 10^{-4}$ |
| **OFA** | | | | | | | | | | | | | |
| Mean | $5.85 \times 10^{-2}$ | $5.67 \times 10^{-2}$ | $5.27 \times 10^{-2}$ | $5.31 \times 10^{-2}$ | $9.00 \times 10^{-1}$ | $7.67 \times 10^{-1}$ | $1.80$ | $4.65 \times 10^{-2}$ | $9.05 \times 10^{-2}$ | $1.66 \times 10^{-1}$ | $4.66 \times 10^{-1}$ | $1.24$ | $4.75 \times 10^{-1}$ |
| 1st Quartile | $4.61 \times 10^{-2}$ | $4.96 \times 10^{-2}$ | $4.59 \times 10^{-2}$ | $4.64 \times 10^{-2}$ | $5.88 \times 10^{-2}$ | $5.58 \times 10^{-2}$ | $5.62 \times 10^{-2}$ | $4.14 \times 10^{-2}$ | $5.10 \times 10^{-2}$ | $5.07 \times 10^{-2}$ | $5.59 \times 10^{-2}$ | $6.37 \times 10^{-2}$ | $5.18 \times 10^{-2}$ |
| 2nd Quartile | $5.91 \times 10^{-2}$ | $5.69 \times 10^{-2}$ | $5.36 \times 10^{-2}$ | $5.18 \times 10^{-2}$ | $1.03 \times 10^{-1}$ | $7.27 \times 10^{-2}$ | $8.48 \times 10^{-1}$ | $4.56 \times 10^{-2}$ | $5.76 \times 10^{-2}$ | $6.21 \times 10^{-2}$ | $8.95 \times 10^{-2}$ | $9.57 \times 10^{-2}$ | $6.94 \times 10^{-2}$ |
| 3rd Quartile | $6.99 \times 10^{-2}$ | $6.51 \times 10^{-2}$ | $5.89 \times 10^{-2}$ | $5.99 \times 10^{-2}$ | $3.07 \times 10^{-1}$ | $1.83 \times 10^{-1}$ | $3.30 \times 10^{-1}$ | $5.15 \times 10^{-2}$ | $6.63 \times 10^{-2}$ | $9.03 \times 10^{-2}$ | $1.62 \times 10^{-1}$ | $2.13 \times 10^{-1}$ | $1.38 \times 10^{-1}$ |
| **WSL** | | | | | | | | | | | | | |
| Mean | $6.33 \times 10^{-3}$ | $1.19 \times 10^{-2}$ | $3.37 \times 10^{-2}$ | $6.80 \times 10^{-2}$ | $2.10 \times 10^{-3}$ | $6.20 \times 10^{-2}$ | $1.48 \times 10^{-2}$ | $2.32 \times 10^{-1}$ | $2.30 \times 10^{-1}$ | $1.92 \times 10^{-1}$ | $2.70 \times 10^{-1}$ | $4.10 \times 10^{-1}$ | $1.40 \times 10^{-1}$ |
| 1st Quartile | $3.93 \times 10^{-4}$ | $5.20 \times 10^{-3}$ | $1.17 \times 10^{-2}$ | $2.62 \times 10^{-2}$ | $7.76 \times 10^{-3}$ | $1.53 \times 10^{-2}$ | $5.72 \times 10^{-2}$ | $1.58 \times 10^{-1}$ | $6.13 \times 10^{-2}$ | $7.05 \times 10^{-2}$ | $1.23 \times 10^{-1}$ | $1.41 \times 10^{-1}$ | $4.80 \times 10^{-2}$ |
| 2nd Quartile | $1.55 \times 10^{-3}$ | $8.08 \times 10^{-3}$ | $2.47 \times 10^{-2}$ | $4.58 \times 10^{-2}$ | $1.47 \times 10^{-2}$ | $5.21 \times 10^{-2}$ | $8.81 \times 10^{-2}$ | $1.37 \times 10^{-1}$ | $1.30 \times 10^{-1}$ | $1.42 \times 10^{-1}$ | $2.52 \times 10^{-1}$ | $2.69 \times 10^{-1}$ | $9.71 \times 10^{-2}$ |
| 3rd Quartile | $5.09 \times 10^{-3}$ | $1.55 \times 10^{-2}$ | $4.56 \times 10^{-2}$ | $8.15 \times 10^{-2}$ | $3.15 \times 10^{-2}$ | $9.59 \times 10^{-2}$ | $1.79 \times 10^{-1}$ | $2.54 \times 10^{-1}$ | $1.95 \times 10^{-1}$ | $1.95 \times 10^{-1}$ | $3.79 \times 10^{-1}$ | $5.05 \times 10^{-1}$ | $1.69 \times 10^{-1}$ |
| **GELS** | | | | | | | | | | | | | |
| Mean | $2.80 \times 10^{-1}$ | $2.11 \times 10^{-1}$ | $1.28 \times 10^{-1}$ | $1.04 \times 10^{-1}$ | $1.81 \times 10^{-1}$ | $1.65 \times 10^{-1}$ | $1.37 \times 10^{-1}$ | $1.10 \times 10^{-1}$ | $1.30 \times 10^{-1}$ | $3.18 \times 10^{-1}$ | $2.79 \times 10^{-1}$ | $5.74 \times 10^{-2}$ | $1.75 \times 10^{-1}$ |
| 1st Quartile | $1.53 \times 10^{-1}$ | $1.14 \times 10^{-1}$ | $7.55 \times 10^{-2}$ | $6.24 \times 10^{-2}$ | $7.95 \times 10^{-2}$ | $7.40 \times 10^{-2}$ | $6.76 \times 10^{-2}$ | $5.64 \times 10^{-2}$ | $1.01 \times 10^{-1}$ | $2.09 \times 10^{-1}$ | $3.06 \times 10^{-2}$ | $2.85 \times 10^{-2}$ | $8.75 \times 10^{-2}$ |
| 2nd Quartile | $2.02 \times 10^{-1}$ | $1.73 \times 10^{-1}$ | $1.01 \times 10^{-1}$ | $7.45 \times 10^{-2}$ | $1.41 \times 10^{-1}$ | $1.20 \times 10^{-1}$ | $9.04 \times 10^{-2}$ | $7.20 \times 10^{-2}$ | $1.33 \times 10^{-1}$ | $3.33 \times 10^{-1}$ | $4.99 \times 10^{-1}$ | $3.97 \times 10^{-2}$ | $1.27 \times 10^{-1}$ |
| 3rd Quartile | $3.03 \times 10^{-1}$ | $2.73 \times 10^{-1}$ | $1.55 \times 10^{-1}$ | $1.15 \times 10^{-1}$ | $2.38 \times 10^{-1}$ | $2.41 \times 10^{-1}$ | $1.27 \times 10^{-1}$ | $1.39 \times 10^{-1}$ | $1.60 \times 10^{-1}$ | $3.95 \times 10^{-1}$ | $2.33 \times 10^{-1}$ | $6.49 \times 10^{-2}$ | $2.04 \times 10^{-1}$ |
| **ARM** | | | | | | | | | | | | | |
| Mean | $1.26 \times 10^{-1}$ | $9.30 \times 10^{-2}$ | $6.68 \times 10^{-2}$ | $6.45 \times 10^{-2}$ | $1.70 \times 10^{-1}$ | $1.41 \times 10^{-1}$ | $1.23 \times 10^{-1}$ | $5.94 \times 10^{-2}$ | $5.06 \times 10^{-2}$ | $4.64 \times 10^{-2}$ | $4.82 \times 10^{-2}$ | $5.23 \times 10^{-2}$ | $8.69 \times 10^{-2}$ |
| 1st Quartile | $6.85 \times 10^{-2}$ | $4.78 \times 10^{-2}$ | $3.78 \times 10^{-2}$ | $3.22 \times 10^{-2}$ | $7.25 \times 10^{-2}$ | $8.41 \times 10^{-2}$ | $7.84 \times 10^{-2}$ | $3.18 \times 10^{-2}$ | $3.18 \times 10^{-2}$ | $3.12 \times 10^{-2}$ | $3.39 \times 10^{-2}$ | $4.02 \times 10^{-2}$ | $4.92 \times 10^{-2}$ |
| 2nd Quartile | $9.96 \times 10^{-2}$ | $6.47 \times 10^{-2}$ | $4.48 \times 10^{-2}$ | $4.36 \times 10^{-2}$ | $1.16 \times 10^{-1}$ | $1.08 \times 10^{-1}$ | $9.55 \times 10^{-2}$ | $4.09 \times 10^{-2}$ | $4.43 \times 10^{-2}$ | $4.17 \times 10^{-2}$ | $4.29 \times 10^{-2}$ | $4.67 \times 10^{-2}$ | $6.57 \times 10^{-2}$ |
| 3rd Quartile | $1.40 \times 10^{-1}$ | $9.12 \times 10^{-2}$ | $7.38 \times 10^{-2}$ | $8.65 \times 10^{-2}$ | $1.72 \times 10^{-1}$ | $1.63 \times 10^{-1}$ | $1.42 \times 10^{-1}$ | $6.22 \times 10^{-2}$ | $6.25 \times 10^{-2}$ | $5.81 \times 10^{-2}$ | $5.49 \times 10^{-2}$ | $6.38 \times 10^{-2}$ | $9.75 \times 10^{-2}$ |
| **WeightedSHAP** | | | | | | | | | | | | | |
| Mean | $2.19 \times 10^{-3}$ | $7.74 \times 10^{-3}$ | $1.81 \times 10^{-2}$ | $2.83 \times 10^{-2}$ | $8.50 \times 10^{-3}$ | $3.02 \times 10^{-2}$ | $8.27 \times 10^{-2}$ | $8.75 \times 10^{-2}$ | $9.45 \times 10^{-2}$ | $1.06 \times 10^{-1}$ | $1.03 \times 10^{-1}$ | $2.25 \times 10^{-1}$ | $6.62 \times 10^{-2}$ |
| 1st Quartile | $2.40 \times 10^{-4}$ | $3.04 \times 10^{-3}$ | $6.62 \times 10^{-3}$ | $8.16 \times 10^{-3}$ | $4.10 \times 10^{-3}$ | $8.39 \times 10^{-3}$ | $2.47 \times 10^{-2}$ | $3.21 \times 10^{-2}$ | $2.53 \times 10^{-2}$ | $2.53 \times 10^{-2}$ | $3.61 \times 10^{-2}$ | $6.98 \times 10^{-2}$ | $2.03 \times 10^{-2}$ |
| 2nd Quartile | $8.74 \times 10^{-4}$ | $4.91 \times 10^{-3}$ | $1.11 \times 10^{-2}$ | $2.12 \times 10^{-2}$ | $7.68 \times 10^{-3}$ | $2.51 \times 10^{-2}$ | $4.33 \times 10^{-2}$ | $5.43 \times 10^{-2}$ | $7.04 \times 10^{-2}$ | $6.18 \times 10^{-2}$ | $6.52 \times 10^{-2}$ | $1.31 \times 10^{-1}$ | $4.14 \times 10^{-2}$ |
| 3rd Quartile | $2.49 \times 10^{-3}$ | $9.36 \times 10^{-3}$ | $1.78 \times 10^{-2}$ | $4.14 \times 10^{-2}$ | $1.27 \times 10^{-2}$ | $3.98 \times 10^{-2}$ | $1.07 \times 10^{-1}$ | $1.10 \times 10^{-1}$ | $1.07 \times 10^{-1}$ | $1.56 \times 10^{-1}$ | $1.26 \times 10^{-1}$ | $2.81 \times 10^{-1}$ | $8.43 \times 10^{-2}$ |

exact enumeration is infeasible, we use a random forest model and compute the true probabilistic values using the algorithm described in Appendix C. (Please see Appendix G for a summary of the datasets in our experiments.) We emphasize that our method for computing the probabilistic values of trees enables the first experiments on medium to large datasets where we can compare the estimates

to the *ground truth* probabilistic values. Such experiments were previously done for Shapley and Banzhaf values [MW25, LWK$^+$25], but not for other probabilistic values.

**Baselines** We compare Linear MSR and Tree MSR to a wide variety of probabilistic value estimators from prior work. For the popular task of estimating Shapley values, we focus on the most effective estimators for general value functions i.e., Permutation SHAP [CGT09], Kernel SHAP [LL17, CL21], and Leverage SHAP [MW25]. We use the optimized implementations of Permutation SHAP and Kernel SHAP in the ubiquitous SHAP library for parity [LL17]. For estimating probabilistic values, there has been substantial recent interest in designing estimators [KZ22a, KZ22b, LZL$^+$22, WJ23, KBMH24, LY24a, LY24b]. These estimators generally use the standard Monte Carlo approach, apply the Maximum Sample Reuse idea, or extend linear regression-based methods. We provide a description of each in Appendix D.

**Error and Uncertainty** We measure the error between the true probabilistic values $\phi$ and the estimated probabilistic values $\tilde{\phi}$ with the $\ell_2$-norm error $\|\phi - \tilde{\phi}\|_2^2 / \|\phi\|_2^2$. All of our tables and figures report summary statistics over at least 10 runs. In the tables, we report the mean, first quartile, median, and third quartiles of the error. (We do not report $+/-$ standard deviation because these are often negative for the small errors in our experiments.) In the figures, we report the mean error.

**Implementation Details** We use scikit-learn [PVG$^+$11] and XGBoost [CG16] for training our models. For the implementations of Permutation SHAP and Kernel SHAP, we use the SHAP library [LL17]. Please see Appendix G for details on how we accessed each dataset. All of our experiments are run on a machine with an Apple M2 chip and 8GB RAM.

We first describe our experiments on the popular task of estimating Shapley values. Figure 2 shows estimator performance by sample size, and Table 1 highlights the corresponding uncertainty statistics when each estimator is run with $m = 40n$ samples. We find that Linear MSR generally improves the prior state-of-the-art Leverage SHAP by making its estimates unbiased, but the gain is marginal. In contrast, the performance Tree MSR depends on how well the tree-based approximation $f$ fits the underlying value function $v$. When the number of samples is smaller, Tree MSR is comparable to prior Shapley value estimators; however, as the number of samples grows, the tree-based model becomes more accurate and Tree MSR often gives the best performance, sometimes by orders of magnitude. As can be seen in Table 1, **Tree MSR can give average error that is** $2.6\times$ **lower than the prior state-of-the-art Leverage SHAP**, when $m = 40n$. For larger sample sizes, Tree MSR outperforms Leverage SHAP by an even wider margin, as shown in Figure 2.

Beyond estimating Shapley values, we run experiments on estimating the more general beta Shapley values and weighted Banzhaf values (see Appendix B for definitions). Figure 6 shows estimator performance by sample size for small datasets (where we can feasibly compute the true probabilistic values of neural networks), and Table 3 highlights the corresponding uncertainty statistics when each estimator is run with $m = 40n$ samples. We present the analogous results for smaller datasets (where we can exactly compute the probabilistic values of neural networks) in Figure 6 and Table 3 in Appendix E. We find that Linear MSR generally plateaus as the number of samples increases. In contrast, Tree MSR gives the best performance across the board, with the gap to the next best estimator widening with the number of samples. We confirm this finding in Figure 4. As can be seen in Table 3, **Tree MSR can give average error that is** $215\times$ **lower than the best probabilistic value estimator from prior work**.

We provide additional experiments on the effect of noisy access to the value function in Appendix F. Figures 7 and 8 suggest that Tree MSR is particularly resilient to noise.

## Limitations and Broader Impacts

The performance of our methods depends on the underlying fit of the learned model. When the dataset is structured or the number of samples is large, the learned model is accurate and the estimators are, too. However, for datasets with less structure or in sample-constrained settings, the performance of our estimators can worsen.

The primary application of our work is in explainable AI, where we seek to understand how features and data points contribute to the performance of machine learning models. We expect the broader impact of our work to be better model explanations, and we do not see substantial negative risks as a result of our research.

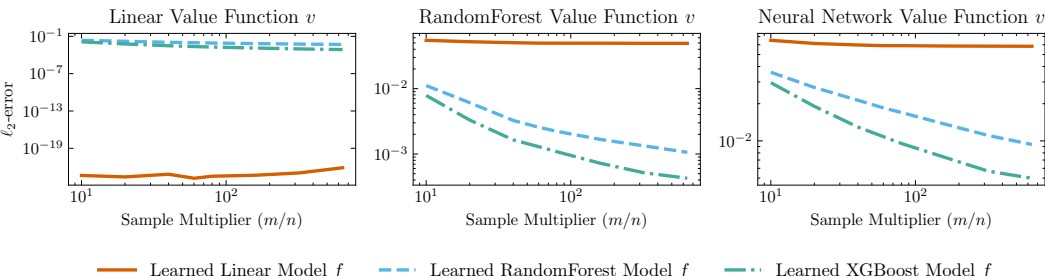

Figure 4: Generalization error between value function $v$ and learned model $f$ by sample size, averaged over all datasets. When the base model is linear, the learned linear model quickly fits it to machine precision. When the base model is a random forest or a neural network, the error of the linear model plateaus while the random forest and XGBoost learned models continue to improve. This phenomenon is reflected in Figures 3 and 6; the performance of Tree MSR continues to improve with the number of samples while Linear MSR plateaus.

## Acknowledgments and Disclosure of Funding

Witter was supported by NSF Graduate Research Fellowship Grant No. DGE-2234660. Liu was partially supported by NSF Awards IIS-2106888 and the DARPA ASKEM and ARPA-H BDF programs. Musco was partially supported by NSF Award CCF-2045590.

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

# A  Proof of Error Bound

**Theorem 2.1** (Regression-Adjustment Guarantee). *The estimates produced by Algorithm 1 are unbiased estimates of the probabilistic values. Further, let $\epsilon, \delta > 0$, and $f_{\max}$ be the learned function $f^{(\ell)}$ with largest generalization error over $\ell \in [k]$. When run with $m = O(n\frac{1}{\epsilon\delta})$ samples, Algorithm 1 produces estimates that satisfy, with probability $1 - \delta$,*

$$\|\tilde{\phi} - \phi\|_2^2 \le \epsilon \sum_{S \subseteq [n]} [v(S) - f_{\max}(S)]^2 \frac{p_{|S|}^2(1 - \frac{|S|}{n}) + p_{|S|-1}^2 \frac{|S|}{n}}{\mathcal{D}(S)}. \tag{6}$$

*Proof of Theorem 2.1.* We will analyze the variance when the samples are drawn with replacement. By Theorem 4 in [Hoe63], the variance can only decrease when samples are drawn without replacement.

Consider $\ell \in [k]$. For simplicity, suppose that $|\mathcal{S}^{(\ell)}| = m/k$. We will first show that each estimated probabilistic value $\tilde{\phi}_i^{(\ell)}$ is unbiased:

$$\mathbb{E}[\tilde{\phi}_i^{(\ell)}] = \phi_i(f^{(\ell)}) + \frac{k}{m}\mathbb{E}\left[\sum_{S' \in \mathcal{S}^{(\ell)}} \sum_{S \subseteq [n]} \frac{\mathbb{1}[S = S']}{\mathcal{D}(S)}[v(S) - f^{(\ell)}(S)]\left(p_{|S|-1}\mathbb{1}[i \in S] - p_{|S|}\mathbb{1}[i \notin S]\right)\right]$$

$$= \phi_i(f^{(\ell)}) + \sum_{S \subseteq [n]} [v(S) - f^{(\ell)}(S)](p_{|S|-1}\mathbb{1}[i \in S] - p_{|S|}\mathbb{1}[i \notin S])$$

$$= \phi_i(f^{(\ell)}) + \sum_{S \subseteq [n]\setminus\{i\}} [v(S \cup \{i\}) - v(S)]p_{|S|} - \sum_{S \subseteq [n]\setminus\{i\}} [f^{(\ell)}(S \cup \{i\}) - f^{(\ell)}(S)]p_{|S|}$$

$$= \sum_{S \subseteq [n]\setminus\{i\}} [v(S \cup \{i\}) - v(S)]p_{|S|} = \phi_i.$$

Since $\tilde{\phi}_i^{(\ell)}$ is unbiased, the final estimate $\mathbb{E}[\frac{1}{k}\sum_{\ell=1}^{k} \tilde{\phi}_i^{(\ell)}]$ is also unbiased by the linearity of expectation. Next, we will analyze the variance of each estimate:

$$\text{Var}[\tilde{\phi}_i^{(\ell)}] = \frac{k^2}{m^2}\text{Var}\left[\sum_{S' \in \mathcal{S}^{(\ell)}} \sum_{S \subseteq [n]} \mathbb{1}[S = S'][v(S) - f^{(\ell)}(S)]\frac{p_{|S|-1}\mathbb{1}[i \in S] - p_{|S|}\mathbb{1}[i \notin S]}{\mathcal{D}(S)}\right]$$

$$\le \frac{k}{m}\sum_{S \subseteq [n]} \mathcal{D}(S)[v(S) - f^{(\ell)}(S)]^2 \left(\frac{p_{|S|-1}\mathbb{1}[i \in S] - p_{|S|}\mathbb{1}[i \notin S]}{\mathcal{D}(S)}\right)^2$$

$$= \frac{k}{m}\sum_{S \subseteq [n]} [v(S) - f^{(\ell)}(S)]^2 \frac{p_{|S|-1}^2\mathbb{1}[i \in S] + p_{|S|}^2\mathbb{1}[i \notin S]}{\mathcal{D}(S)}. \tag{7}$$

Let $\phi \in \mathbb{R}^n$ and $\tilde{\phi}^{(\ell)} \in \mathbb{R}^n$ be vectors containing the true and estimated probabilistic values, respectively. We will analyze the random variable $\|\phi - \tilde{\phi}^{(\ell)}\|^2$. By linearity of expectation, we have

$$\mathbb{E}[\|\phi - \tilde{\phi}^{(\ell)}\|^2] = \mathbb{E}\left[\sum_{i=1}^{n} (\phi_i - \tilde{\phi}_i^{(\ell)})^2\right] = \sum_{i=1}^{n} \mathbb{E}\left[(\phi_i - \tilde{\phi}_i^{(\ell)})^2\right] = \sum_{i=1}^{n} \text{Var}[\phi_i - \tilde{\phi}_i^{(\ell)}] = \sum_{i=1}^{n} \text{Var}[\tilde{\phi}_i^{(\ell)}]$$

where the penultimate equality follows because $\mathbb{E}[\tilde{\phi}_i^{(\ell)}] = \phi_i$ and the final equality follows because $\phi_i$ is a constant with respect to the randomness of the estimator. Plugging in Equation (7), we get

$$\mathbb{E}[\|\phi - \tilde{\phi}^{(\ell)}\|^2] \le \sum_{i=1}^{n} \frac{k}{m}\sum_{S \subseteq [n]} [v(S) - f^{(\ell)}(S)]^2 \frac{p_{|S|-1}^2\mathbb{1}[i \in S] + p_{|S|}^2\mathbb{1}[i \notin S]}{\mathcal{D}(S)}$$

$$= \frac{k}{m}\sum_{S \subseteq [n]} [v(S) - f^{(\ell)}(S)]^2 \frac{p_{|S|-1}^2|S| + p_{|S|}^2(n - |S|)}{\mathcal{D}(S)}$$

$$\le \frac{k}{m}\sum_{S \subseteq [n]} [v(S) - f_{\max}(S)]^2 \frac{p_{|S|-1}^2|S| + p_{|S|}^2(n - |S|)}{\mathcal{D}(S)} \tag{8}$$

where

$$f_{\max} := f^{(\ell^*)}, \quad \text{where } \ell^* = \arg\max_{\ell \in [k]} \sum_{S \subseteq [n]} [v(S) - f^{(\ell)}(S)]^2 \frac{p_{|S|-1}^2 |S| + p_{|S|}^2 (n - |S|)}{\mathcal{D}(S)}.$$

We now apply Markov's inequality to $\|\phi - \tilde{\phi}^{(\ell)}\|^2$ for each $\ell$:

$$\Pr\left(\|\phi - \tilde{\phi}^{(\ell)}\|^2 \geq \frac{1}{\delta'}\mathbb{E}[\|\phi - \tilde{\phi}^{(\ell)}\|^2]\right) \leq \delta'.$$

We are interested in the final estimate $\tilde{\phi} = \frac{1}{k}\sum_{\ell=1}^k \tilde{\phi}^{(\ell)}$. By the Union Bound, we have, with probability at most $k\delta'$,

$$\sum_{\ell=1}^k \|\phi - \tilde{\phi}^{(\ell)}\| \geq \sum_{\ell=1}^k \sqrt{\frac{1}{\delta'}\mathbb{E}[\|\phi - \tilde{\phi}^{(\ell)}\|^2]}.$$

By the triangle inequality, setting $\delta' = \delta/k$, and taking the complement, we have, with probability $1 - \delta$,

$$k\|\phi - \tilde{\phi}\| \leq \sum_{\ell=1}^k \|\phi - \tilde{\phi}^{(\ell)}\| \qquad\qquad \text{(by triangle inequality)}$$

$$\leq \sum_{\ell=1}^k \sqrt{\frac{k}{\delta}\mathbb{E}[\|\phi - \tilde{\phi}^{(\ell)}\|^2]} \qquad\qquad \text{(by setting } \delta' = \delta/k\text{)}$$

$$\leq k\sqrt{\frac{k}{\delta}\frac{k}{m}\sum_{S \subseteq [n]} [v(S) - f_{\max}(S)]^2 \frac{p_{|S|-1}^2 |S| + p_{|S|}^2 (n - |S|)}{\mathcal{D}(S)}}. \quad \text{(by Equation (8))}$$

Then, with probability $1 - \delta$,

$$\|\phi - \tilde{\phi}\|^2 \leq \frac{k}{\delta}\frac{k}{m}\sum_{S \subseteq [n]} [v(S) - f_{\max}(S)]^2 \frac{p_{|S|-1}^2 |S| + p_{|S|}^2 (n - |S|)}{\mathcal{D}(S)}.$$

The theorem statement follows by setting $m = k^2\frac{n}{\epsilon\delta} = O(\frac{n}{\epsilon\delta})$. $\qquad\qquad\qquad\qquad\square$

# B   Beta Shapley and Weighted Banzhaf Values

Probabilistic values satisfy three intuitive properties:

- Linearity: The probabilistic value of a linear combination of value functions is the linear combination of the probabilistic values for each value function i.e., $\phi_i(av + bw) = a\phi_i(v) + b\phi_i(w)$ for real values $a, b \in \mathbb{R}$ and games $v, w : 2^{[n]} \to \mathbb{R}$.

- Null Player: The probabilistic value for a player that has no effect on any coalition is 0 i.e., if $v(S \cup \{i\}) = v(S)$ for all $S$ then $\phi_i = 0$.

- Symmetry: If two players contribute equally to all coalitions then they have the same probabilistic value i.e., if $v(S \cup \{i\}) = v(S \cup \{j\})$ for all $S$ then $\phi_i = \phi_j$.

In addition to these three properties, Shapley and Banzhaf values satisfy the efficiency and 2-efficiency properties, respectively:

- Efficiency: The sum of probabilistic values is the difference between the whole coalition and the empty set i.e., $\sum_{i=1}^{n} \phi_i = v([n]) - v(\emptyset)$. Efficiency is desirable in settings where we want to attribute the value $v([n]) - v(\emptyset)$ to each player e.g., model prediction explanation or cost-sharing.

- 2-Efficiency: Let $v'$ be a game where $i$ and $j$ are combined into a single player $(i, j)$. That is, $v'$ is defined on $n - 1$ players where the $i$ and $j$ players in $v$ are always considered together; effectively, $v'$ is only defined on subsets that contain both $i$ and $j$ or contain neither $i$ nor $j$. The 2-efficiency property requires that $\phi_i(v) + \phi_j(v) = \phi_{(i,j)}(v')$ for all $i, j$. 2-Efficiency is desirable in settings where players can be combined into subgroups e.g., federated learning with client aggregation or games with alliances between players.

Shapley values are the most popular probabilistic value, in part because they are the only probabilistic value to satisfy the efficiency property [Sha51]. The probabilistic weights for Shapley values are $p_{|S|} = \frac{1}{n}\binom{n-1}{|S|}^{-1}$, which weights all *set sizes* equally.

The efficiency property is useful in settings where we want to allocate the total value of the game to the players. However, efficiency is not always appropriate especially when there are non-linear interactions between players that cannot be attributed to individuals. For these cases, a more appropriate property may be 2-efficiency which requires that probabilistic values add if players are merged. The only probabilistic value that satisfies 2-efficiency is the Banzhaf value [Pen46, BI64]. The probabilistic weight for Banzhaf values is $p_{|S|} = 1/2^{n-1}$, which weights all *sets* equally.

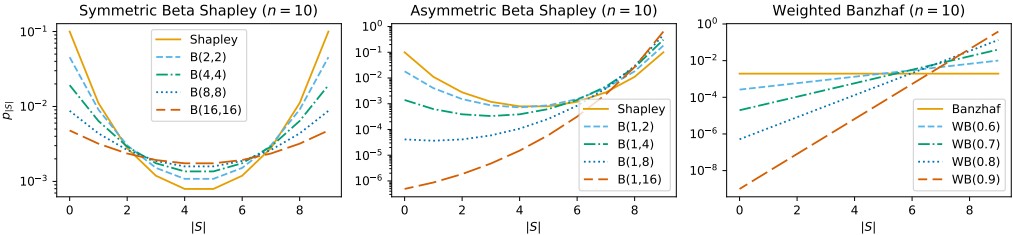

Figure 5: Probabilistic values by subset size for $n = 10$. Beta Shapley values $B(\alpha, \beta)$ generalize Shapley values for $\alpha, \beta \in [1, \infty)$; increasing both $\alpha$ and $\beta$ flattens beta Shapley values while increasing just $\alpha$ (or just $\beta$) tilts beta Shapley values. Weighted Banzhaf values $WB(p)$ generalize Banzhaf values for $p \in (0, 1)$; increasing (or decreasing) $p$ tilts weighted Banzhaf values.

Both Shapley and Banzhaf values have been generalized to beta Shapley values [KZ22a] and weighted Banzhaf values [LY24c], respectively. Figure 5 plots $p_{|S|}$ for various beta Shapley and weighted Banzhaf values when $n = 10$. Beta Shapley values are defined by two parameters $\alpha, \beta \in [1, \infty)$. In particular, the probabilistic weight is

$$\frac{\mathrm{Beta}(|S| + \beta, n - |S| - 1 + \alpha)}{\mathrm{Beta}(\alpha, \beta)}.$$

Setting $\alpha = \beta = 1$ recovers Shapley values. Weighted Banzhaf values are defined by one parameter $p \in (0, 1)$. In particular, the probabilistic weight is

$$p^{|S|}(1-p)^{n-|S|-1}.$$

Setting $p = \frac{1}{2}$ recovers Banzhaf values.

## C  Computing Probabilistic Values of Tree-based Models

In this section, we show how to efficiently compute the probabilistic values of a tree when the value function is interventional feature attribution. Unfortunately, we cannot directly generalize analogous algorithms for Shapley and Banzhaf values [LL17, LEC$^+$20, KMM$^+$22], since they uses a property which does not hold for all probabilistic values. In particular, prior approaches assume that if a value function only has contributions from $n' < n$ players, the Shapley/Banzhaf value on the induced game of those $n'$ players is the same as the Shapley/Banzhaf value on the original value function with all $n$ players. Our approach is based on an alternative way of viewing tree-based models that avoids the need for this property.

Consider the value function $v : 2^{[n]} \to \mathbb{R}$ induced by a tree with explicand $\mathbf{x}^e$ and baseline $\mathbf{x}^b$. We decompose $v$ into a sum of *path value functions* $\{v^P\}_P$, where each $v^P : 2^{[n]} \to S$ corresponds to a distinct root-to-leaf path $P$. In particular,

$$v(S) = \sum_P v^P(S), \qquad \text{where} \quad v^P(S) = \begin{cases} \text{leaf value of } P & \text{if } S \text{ follows path } P \text{ on } \mathbf{x}^e, \mathbf{x}^b, \\ 0 & \text{otherwise.} \end{cases}$$

By the linearity property of probabilistic values, the contribution of feature $i$ to the full tree model $v$ can be expressed as the sum of its contributions to each path model $v^P$. Specifically,

$$\phi_i(v) = \phi_i\left(\sum_P v^P\right) = \sum_P \phi_i(v^P).$$

Therefore, it suffices to compute $\phi_i(v^P)$ for each path model $v^P$ and aggregate their contributions over all paths. To this end, we will introduce the following notation. Given probabilistic weights $\mathbf{p} = [p_0, \ldots, p_{n-1}] \in [0,1]^n$, for each path $P$, define

- $S_P$ as the set of features in $P$ whose conditions are satisfied by $\mathbf{x}^e$ but not by $\mathbf{x}^b$,
- $N_P$ as the set of features in $P$ whose conditions are satisfied by $\mathbf{x}^b$ but not by $\mathbf{x}^e$,
- $\ell_P$ as the final leaf value on path $P$.

Recall we can write the $i$th probabilistic value as

$$\phi_i(v) = \sum_{S \subseteq [n] : i \in S} p_{|S|-1} v(S) - \sum_{S \subseteq [n] : i \notin S} p_{|S|} v(S).$$

Using the definition of $v^P$, $S_P$, and $N_P$, we can consider $\phi_i(v^P)$ in three cases:

- **Case 1: $i \in S_P$:** We need $i \in S$ in order to reach the leaf i.e., $v^P(S) = 0$ unless $S_P \subseteq S \subseteq [n] \setminus N_P$ and $i \in S$. Then,

$$\phi_i(v^P) = \sum_{S_P \subseteq S \subseteq [n] \setminus N_P} p_{|S|-1} \cdot \ell_P = \sum_{l=|S_P|}^{n-|N_P|} p_{l-1} \binom{n - |N_P| - |S_P|}{l - |S_P|} \cdot \ell_P$$

- **Case 2: $i \in N_P$:** We need $i \notin S$ in order to reach the leaf i.e., $v^P(S) = 0$ unless $S_P \subseteq S \subseteq [n] \setminus N_P$ and $i \notin S$. Then,

$$\phi_i(v^P) = - \sum_{l=|S_P|}^{n-|N_P|} p_l \binom{n - |N_P| - |S_P|}{l - |S_P|} \cdot \ell_P$$

- **Case 3: $i \notin N_P$ and $i \notin S_P$:** We reach the leaf whether $i \in S$ or not i.e., $v^P(S) = 0$ unless $S_P \subseteq S \subseteq [n] \setminus N_P$. Then,

$$\phi_i(v^P) = \sum_{l=|S_P|+1}^{n-|N_P|} p_{l-1} \binom{n - |N_P| - |S_P| - 1}{l - |S_P| - 1} \ell_P$$

$$- \sum_{l=|S_P|}^{n-|N_P|-1} p_l \binom{n - |N_P| - |S_P| - 1}{l - |S_P|} \ell_P = 0$$

## C.1 TreeProb Pseudocode

Algorithm 2 efficiently explores all root-to-leaf paths, maintaining counters for how many times each feature has been "seen" under the explicand (ef_seen) or the baseline (bf_seen). When a branching feature is encountered for the first time on a path, the algorithm branches into two recursive calls—one following $\mathbf{x}^e$, the other $\mathbf{x}^b$—and updates the *cardinalities* $s_P := |S_P|$ or $n_P := |N_P|$ accordingly, depending on which feature value is consistent with the split. At each leaf node, the algorithm computes the contribution based on the current $(s_P, n_P)$ and aggregates these into the overall attribution vector $\phi$. Algorithm 2 preserves the original complexity and traversal logic of Tree SHAP, while generalizing the feature-contribution calculation to the probabilistic formulation described above, making it applicable to any probabilistic value.

**Algorithm 2** TreeProb with Interventional Feature Perturbation

---

1: **Input:** $n$: number of players; $\mathbf{p} \in [0,1]^n$: probabilistic weights
2: **Output:** Exact probabilistic values $\phi_1, \ldots, \phi_n$

3: **function** RECURSE(node, $s_P$, $n_P$, ef_seen, bf_seen)
4: **if** node is a leaf **then**

5: $\quad$ pos_term $\leftarrow$ node.value $\cdot \displaystyle\sum_{l=s_P}^{n-n_P} p_{l-1} \binom{n - n_P - s_P}{l - s_P}$

6: $\quad$ neg_term $\leftarrow$ $-$node.value $\cdot \displaystyle\sum_{l=s_P}^{n-n_P} p_l \binom{n - n_P - s_P}{l - s_P}$

7: $\quad$ **return** (pos_term, neg_term)
8: **end if**

9: $x_e\_$child $\leftarrow \begin{cases} \text{node.leftchild} & \text{if } x_e[\text{node.feat}] < \text{node.t} \\ \text{node.rightchild} & \text{otherwise} \end{cases}$

10: $x_b\_$child $\leftarrow \begin{cases} \text{node.leftchild} & \text{if } x_b[\text{node.feat}] < \text{node.t} \\ \text{node.rightchild} & \text{otherwise} \end{cases}$

11: **if** ef_seen[node.feat] $> 0$ **then**
12: $\quad$ **return** RECURSE($x_e\_$child, $s_P$, $n_P$, ef_seen, bf_seen)
13: **end if**
14: **if** bf_seen[node.feat] $> 0$ **then**
15: $\quad$ **return** RECURSE($x_b\_$child, $s_P$, $n_P$, ef_seen, bf_seen)
16: **end if**
17: **if** $x_e\_$child $= x_b\_$child **then**
18: $\quad$ **return** RECURSE($x_e\_$child, $s_P$, $n_P$, ef_seen, bf_seen)
19: **else**
20: $\quad$ ef_seen[node.feat] $\leftarrow$ ef_seen[node.feat] $+ 1$
21: $\quad$ (pos$_e$, neg$_e$) $\leftarrow$ RECURSE($x_e\_$child, $s_P + 1$, $n_P$, ef_seen, bf_seen)
22: $\quad$ ef_seen[node.feat] $\leftarrow$ ef_seen[node.feat] $- 1$
23: $\quad$ bf_seen[node.feat] $\leftarrow$ bf_seen[node.feat] $+ 1$
24: $\quad$ (pos$_b$, neg$_b$) $\leftarrow$ RECURSE($x_b\_$child, $s_P$, $n_P + 1$, ef_seen, bf_seen)
25: $\quad$ bf_seen[node.feat] $\leftarrow$ bf_seen[node.feat] $- 1$
26: $\quad$ $\phi_{\text{temp}}$[node.feat] $\leftarrow$ $\phi_{\text{temp}}$[node.feat] $+$ (pos$_e$ + neg$_b$)
27: $\quad$ **return** (pos$_e$ + pos$_b$, neg$_e$ + neg$_b$)
28: **end if**
29: **end function**

30: **Initialize** $\phi \leftarrow \mathbf{0}^n$
31: **for** each tree $t$ in the ensemble **do**
32: $\quad$ **for** each baseline $\mathbf{x}^b$ in baselines **do**
33: $\quad\quad$ $\phi_{\text{temp}} \leftarrow \mathbf{0}^n$
34: $\quad\quad$ RECURSE($t$.root, 0, 0, $\mathbf{0}^n$, $\mathbf{0}^n$)
35: $\quad\quad$ $\phi \leftarrow \phi + \phi_{\text{temp}}$
36: $\quad$ **end for**
37: **end for**
38: **return** $\phi/$(number of trees $\times$ number of baselines)

---

# D  Baselines

In this section, we describe the baselines we compare against for estimating probabilistic values. These baselines broadly fall into three categories: standard Monte Carlo methods, maximum sample reuse methods, and regression methods.

For a more technical description of many of these baselines, please refer to Appendix D of [LY24b].

**Monte Carlo Methods** The standard Monte Carlo estimator estimates each probabilistic value individually by sampling each term in the summation with probability proportional to its weight.

*Weighted Sampling Lift (WSL)* [KZ22a] is the standard Monte Carlo, but where subsets are sampled according to the Shapley weights and reweighted to produce unbiased estimates of the probabilistic values.

*Permutation SHAP* [CGT09] is similar to Monte Carlo estimates except that subsets are sampled in permutations; that is, the first element in the sampled permutation is one sampled subset, the first two elements are another sampled subset, and so on. Because each set *size* is weighted equally by Shapley values, sampling permutations without any reweighting gives estimates that are the Shapley values in expectation. Permutation SHAP gives close to state-of-the-art performance.

*Weighted SHAP* estimator [KZ22b] generalizes permutation sampling approach to probabilistic value. Random permutations are drawn as before but now reweighted by the probabilistic value weights so that the final estimates are unbiased.

**Maximum Sample Reuse Methods** Monte Carlo methods are unbiased but inefficient in the sense that they only use each sample to compute the estimates of one or two probabilistic values. The key observation of Maximum Sample Reuse (MSR) estimators is that each probabilistic value $\phi_i$ can be written as two summations, one over sets that include $i$ and one over sets that do not. Then the MSR methods use each sample to update every summation. First used for Banzhaf values [WJ23], MSR has since been generalized to other probabilistic values with different sampling distributions.

*Approximation without Requesting Marginals (ARM)* [KBMH24] is a kind of MSR estimator. Half the samples are drawn with probability $p_{|S|-1}$ while the other half are drawn with probability $p_{|S|}$. In order to avoid the numerical instability of reweighting, the final estimate only includes the first half of samples if $i \in S$ and the second if $i \notin S$.

*One sample Fits All (OFA)* [LY24b] similarly uses maximum sample reuse but samples according to a more complicated distribution.

**Regression Methods** A parallel line of work fits linear models $f$ to $v$ then returns the probabilistic values of $f$. These approaches originate for estimating Shapley values, and are based on a linear regression problem that exactly recovers the Shapley values when solved exactly [CGKR88].

*Kernel SHAP* [LL17] samples subsets from this regression problem with probability proportional to their weighting in the regression problem.

*Leverage SHAP* [MW25] similarly samples subsets but with probability proportional to their *statistical leverage* in the regression problem, resulting in state-of-the-art Shapley value estimates and error bounds that depend on the fit of $f$ to $v$.

*Kernel Banzhaf* [LWK$^+$25] is similar to Kernel SHAP and Leverage SHAP but estimates Banzhaf values, and is based on a regression formulation specific to Banzhaf values [HH92].

There is a known generalization of the Shapley value regression problem [RVZ98], which, when solved exactly, recovers probabilistic values up to additive constant. Since Shapley values satisfy efficiency, this constant is efficient to exactly compute for Shapley values. However, for general probabilistic values, the constant depends on the entire value function $v$ and must be estimated, introducing another source of error.

The *Generic Estimator based on Least Squares (GELS)* [LY24a] estimates the constant by adding a dummy variable with probabilistic value 0. Instead of fitting a linear function $f$, GELS considers the closed-form solution to the regression problem and effectively applies a maximum sample reuse estimator to the underlying matrix-vector multiplication. The final estimates are adjusted by subtracting the value of the dummy variable.

The *Average Marginal Effect (AME)* [LZL+22] is another regression estimator that uses a different regression formulation. For probabilistic values that satisfy a specific condition, the probabilistic values can be written as an infinitely tall regression problem. The estimator samples this regression problem and solves the approximate version.

# E   Experiments on Small Datasets with Neural Network Models

Table 3: Summary statistics of the $\ell_2$-norm error between estimated and true probabilistic values when $m = 40n$. We summarize the error over small datasets ($n < 30$), for which the probabilistic values of a neural network model can be feasibly computed. On average over all probabilistic values, Tree MSR produces estimates with mean error that is $150\times$ lower than the best estimator from prior work.

| | B(1,1) | B(2,2) | B(4,4) | B(8,8) | B(1,2) | B(1,4) | B(1,8) | WB(0.5) | WB(0.6) | WB(0.7) | WB(0.8) | WB(0.9) | Mean |
|---|---|---|---|---|---|---|---|---|---|---|---|---|---|
| **LinearMSR** | | | | | | | | | | | | | |
| Mean | $9.41 \times 10^{-4}$ | $1.42 \times 10^{-3}$ | $1.14 \times 10^{-3}$ | $1.30 \times 10^{-3}$ | $1.41 \times 10^{-2}$ | $2.16 \times 10^{-2}$ | $2.74 \times 10^{-2}$ | $1.29 \times 10^{-3}$ | $6.80 \times 10^{-3}$ | $4.67 \times 10^{-3}$ | $5.61 \times 10^{-3}$ | $1.45 \times 10^{-2}$ | $8.39 \times 10^{-3}$ |
| 1st Quartile | $1.41 \times 10^{-7}$ | $1.81 \times 10^{-7}$ | $1.54 \times 10^{-7}$ | $1.03 \times 10^{-7}$ | $5.85 \times 10^{-5}$ | $1.33 \times 10^{-4}$ | $1.56 \times 10^{-4}$ | $1.42 \times 10^{-7}$ | $2.08 \times 10^{-5}$ | $1.70 \times 10^{-5}$ | $1.68 \times 10^{-5}$ | $4.36 \times 10^{-5}$ | $3.72 \times 10^{-5}$ |
| 2nd Quartile | $7.43 \times 10^{-6}$ | $1.31 \times 10^{-5}$ | $1.16 \times 10^{-5}$ | $7.83 \times 10^{-6}$ | $7.26 \times 10^{-4}$ | $1.25 \times 10^{-3}$ | $7.70 \times 10^{-4}$ | $1.08 \times 10^{-5}$ | $1.58 \times 10^{-4}$ | $2.27 \times 10^{-4}$ | $2.85 \times 10^{-4}$ | $4.15 \times 10^{-4}$ | $3.23 \times 10^{-4}$ |
| 3rd Quartile | $7.27 \times 10^{-4}$ | $2.41 \times 10^{-3}$ | $1.97 \times 10^{-3}$ | $1.11 \times 10^{-3}$ | $1.50 \times 10^{-2}$ | $2.60 \times 10^{-2}$ | $4.47 \times 10^{-2}$ | $1.77 \times 10^{-3}$ | $3.71 \times 10^{-3}$ | $4.29 \times 10^{-3}$ | $5.67 \times 10^{-3}$ | $2.43 \times 10^{-2}$ | $1.10 \times 10^{-2}$ |
| **TreeMSR** | | | | | | | | | | | | | |
| Mean | $3.83 \times 10^{-4}$ | $5.36 \times 10^{-4}$ | $5.96 \times 10^{-4}$ | $6.46 \times 10^{-4}$ | $4.58 \times 10^{-4}$ | $3.59 \times 10^{-4}$ | $2.90 \times 10^{-4}$ | $6.61 \times 10^{-4}$ | $4.51 \times 10^{-4}$ | $6.36 \times 10^{-4}$ | $3.50 \times 10^{-4}$ | $1.26 \times 10^{-4}$ | $4.58 \times 10^{-4}$ |
| 1st Quartile | $7.28 \times 10^{-7}$ | $1.08 \times 10^{-6}$ | $1.90 \times 10^{-6}$ | $1.08 \times 10^{-6}$ | $5.35 \times 10^{-7}$ | $2.60 \times 10^{-7}$ | $1.11 \times 10^{-7}$ | $1.09 \times 10^{-6}$ | $7.71 \times 10^{-7}$ | $6.55 \times 10^{-7}$ | $3.69 \times 10^{-7}$ | $4.36 \times 10^{-8}$ | $7.19 \times 10^{-7}$ |
| 2nd Quartile | $2.63 \times 10^{-5}$ | $2.11 \times 10^{-5}$ | $3.08 \times 10^{-5}$ | $3.28 \times 10^{-5}$ | $1.21 \times 10^{-5}$ | $9.25 \times 10^{-6}$ | $7.16 \times 10^{-6}$ | $3.13 \times 10^{-5}$ | $2.13 \times 10^{-5}$ | $1.53 \times 10^{-5}$ | $1.14 \times 10^{-5}$ | $5.51 \times 10^{-6}$ | $1.87 \times 10^{-5}$ |
| 3rd Quartile | $2.16 \times 10^{-4}$ | $2.45 \times 10^{-4}$ | $2.41 \times 10^{-4}$ | $1.69 \times 10^{-4}$ | $1.81 \times 10^{-4}$ | $1.34 \times 10^{-4}$ | $1.57 \times 10^{-4}$ | $1.78 \times 10^{-4}$ | $1.67 \times 10^{-4}$ | $2.44 \times 10^{-4}$ | $1.08 \times 10^{-4}$ | $6.15 \times 10^{-5}$ | $1.75 \times 10^{-4}$ |
| **OFA** | | | | | | | | | | | | | |
| Mean | $3.17 \times 10^{-2}$ | $3.72 \times 10^{-2}$ | $4.79 \times 10^{-2}$ | $4.30 \times 10^{-2}$ | $3.04 \times 10^{-2}$ | $2.08 \times 10^{-2}$ | $1.48 \times 10^{-2}$ | $4.48 \times 10^{-2}$ | $2.18 \times 10^{-1}$ | $1.96 \times 10^{-1}$ | $4.94 \times 10^{-2}$ | $1.55 \times 10^{-1}$ | $7.42 \times 10^{-2}$ |
| 1st Quartile | $2.29 \times 10^{-2}$ | $2.95 \times 10^{-2}$ | $3.27 \times 10^{-2}$ | $3.12 \times 10^{-2}$ | $2.35 \times 10^{-2}$ | $1.47 \times 10^{-2}$ | $1.04 \times 10^{-2}$ | $2.82 \times 10^{-2}$ | $4.67 \times 10^{-2}$ | $3.56 \times 10^{-2}$ | $2.80 \times 10^{-2}$ | $1.51 \times 10^{-2}$ | $2.65 \times 10^{-2}$ |
| 2nd Quartile | $3.04 \times 10^{-2}$ | $3.60 \times 10^{-2}$ | $4.95 \times 10^{-2}$ | $2.97 \times 10^{-2}$ | $2.07 \times 10^{-2}$ | $1.27 \times 10^{-2}$ | $2.07 \times 10^{-2}$ | $4.00 \times 10^{-2}$ | $6.16 \times 10^{-2}$ | $4.90 \times 10^{-2}$ | $3.70 \times 10^{-2}$ | $2.43 \times 10^{-2}$ | $3.60 \times 10^{-2}$ |
| 3rd Quartile | $3.82 \times 10^{-2}$ | $4.33 \times 10^{-2}$ | $5.92 \times 10^{-2}$ | $5.27 \times 10^{-2}$ | $3.70 \times 10^{-2}$ | $2.55 \times 10^{-2}$ | $1.72 \times 10^{-2}$ | $5.59 \times 10^{-2}$ | $7.55 \times 10^{-2}$ | $6.84 \times 10^{-2}$ | $5.05 \times 10^{-2}$ | $6.45 \times 10^{-2}$ | $4.90 \times 10^{-2}$ |
| **WSL** | | | | | | | | | | | | | |
| Mean | $1.07 \times 10^{-2}$ | $3.67 \times 10^{-2}$ | $4.26 \times 10^{-2}$ | $3.34 \times 10^{-1}$ | $5.00 \times 10^{-2}$ | $7.18 \times 10^{-2}$ | $1.31 \times 10^{-1}$ | $5.73 \times 10^{-1}$ | $1.24 \times 10^{-1}$ | $8.35 \times 10^{-2}$ | $1.16 \times 10^{-1}$ | $1.48 \times 10^{-1}$ | $1.48 \times 10^{-1}$ |
| 1st Quartile | $6.31 \times 10^{-5}$ | $2.76 \times 10^{-3}$ | $6.75 \times 10^{-3}$ | $1.76 \times 10^{-2}$ | $8.82 \times 10^{-3}$ | $1.49 \times 10^{-2}$ | $3.40 \times 10^{-2}$ | $2.60 \times 10^{-2}$ | $9.91 \times 10^{-3}$ | $7.67 \times 10^{-3}$ | $2.85 \times 10^{-2}$ | $5.32 \times 10^{-2}$ | $1.75 \times 10^{-2}$ |
| 2nd Quartile | $3.76 \times 10^{-4}$ | $1.13 \times 10^{-2}$ | $2.30 \times 10^{-2}$ | $4.77 \times 10^{-2}$ | $2.38 \times 10^{-2}$ | $3.86 \times 10^{-2}$ | $8.61 \times 10^{-2}$ | $7.16 \times 10^{-2}$ | $5.91 \times 10^{-2}$ | $5.65 \times 10^{-2}$ | $7.62 \times 10^{-2}$ | $1.08 \times 10^{-1}$ | $5.02 \times 10^{-2}$ |
| 3rd Quartile | $1.35 \times 10^{-2}$ | $2.42 \times 10^{-2}$ | $5.95 \times 10^{-2}$ | $9.04 \times 10^{-2}$ | $4.18 \times 10^{-2}$ | $8.59 \times 10^{-2}$ | $1.88 \times 10^{-1}$ | $1.27 \times 10^{-1}$ | $1.29 \times 10^{-1}$ | $1.06 \times 10^{-1}$ | $1.57 \times 10^{-1}$ | $2.30 \times 10^{-1}$ | $1.04 \times 10^{-1}$ |
| **GELS** | | | | | | | | | | | | | |
| Mean | $1.82 \times 10^{-1}$ | $1.16 \times 10^{-1}$ | $1.14 \times 10^{-1}$ | $1.15 \times 10^{-1}$ | $8.83 \times 10^{-2}$ | $8.02 \times 10^{-2}$ | $6.83 \times 10^{-2}$ | $1.20 \times 10^{-1}$ | $7.42 \times 10^{-2}$ | $5.65 \times 10^{-2}$ | $4.82 \times 10^{-2}$ | $5.59 \times 10^{-2}$ | $9.33 \times 10^{-2}$ |
| 1st Quartile | $8.97 \times 10^{-2}$ | $5.91 \times 10^{-2}$ | $6.20 \times 10^{-2}$ | $5.60 \times 10^{-2}$ | $4.79 \times 10^{-2}$ | $3.41 \times 10^{-2}$ | $3.54 \times 10^{-2}$ | $5.94 \times 10^{-2}$ | $3.33 \times 10^{-2}$ | $3.03 \times 10^{-2}$ | $2.88 \times 10^{-2}$ | $3.07 \times 10^{-2}$ | $4.72 \times 10^{-2}$ |
| 2nd Quartile | $1.42 \times 10^{-1}$ | $9.13 \times 10^{-2}$ | $8.73 \times 10^{-2}$ | $8.24 \times 10^{-2}$ | $7.61 \times 10^{-2}$ | $5.63 \times 10^{-2}$ | $5.23 \times 10^{-2}$ | $9.24 \times 10^{-2}$ | $5.07 \times 10^{-2}$ | $4.33 \times 10^{-2}$ | $3.85 \times 10^{-2}$ | $4.27 \times 10^{-2}$ | $7.13 \times 10^{-2}$ |
| 3rd Quartile | $2.26 \times 10^{-1}$ | $1.26 \times 10^{-1}$ | $1.35 \times 10^{-1}$ | $1.45 \times 10^{-1}$ | $1.11 \times 10^{-1}$ | $9.56 \times 10^{-2}$ | $7.34 \times 10^{-2}$ | $1.63 \times 10^{-1}$ | $8.37 \times 10^{-2}$ | $6.92 \times 10^{-2}$ | $6.24 \times 10^{-2}$ | $7.33 \times 10^{-2}$ | $1.14 \times 10^{-1}$ |
| **ARM** | | | | | | | | | | | | | |
| Mean | $2.06 \times 10^{-1}$ | $1.04 \times 10^{-1}$ | $1.68 \times 10^{-1}$ | $1.27 \times 10^{-1}$ | $7.41 \times 10^{-2}$ | $5.13 \times 10^{-2}$ | $4.91 \times 10^{-2}$ | $9.18 \times 10^{-2}$ | $6.13 \times 10^{-2}$ | $4.86 \times 10^{-2}$ | $5.65 \times 10^{-2}$ | $4.04 \times 10^{-2}$ | $8.98 \times 10^{-2}$ |
| 1st Quartile | $4.13 \times 10^{-2}$ | $3.69 \times 10^{-2}$ | $3.32 \times 10^{-2}$ | $2.80 \times 10^{-2}$ | $3.72 \times 10^{-2}$ | $3.36 \times 10^{-2}$ | $2.77 \times 10^{-2}$ | $2.83 \times 10^{-2}$ | $2.80 \times 10^{-2}$ | $3.30 \times 10^{-2}$ | $3.54 \times 10^{-2}$ | $2.20 \times 10^{-2}$ | $3.20 \times 10^{-2}$ |
| 2nd Quartile | $6.79 \times 10^{-2}$ | $6.28 \times 10^{-2}$ | $4.76 \times 10^{-2}$ | $4.57 \times 10^{-2}$ | $5.77 \times 10^{-2}$ | $4.82 \times 10^{-2}$ | $4.28 \times 10^{-2}$ | $3.69 \times 10^{-2}$ | $4.33 \times 10^{-2}$ | $4.01 \times 10^{-2}$ | $4.64 \times 10^{-2}$ | $3.02 \times 10^{-2}$ | $4.75 \times 10^{-2}$ |
| 3rd Quartile | $9.67 \times 10^{-2}$ | $9.79 \times 10^{-2}$ | $8.27 \times 10^{-2}$ | $6.68 \times 10^{-2}$ | $9.99 \times 10^{-2}$ | $5.86 \times 10^{-2}$ | $6.62 \times 10^{-2}$ | $5.70 \times 10^{-2}$ | $7.98 \times 10^{-2}$ | $6.26 \times 10^{-2}$ | $6.33 \times 10^{-2}$ | $4.62 \times 10^{-2}$ | $7.31 \times 10^{-2}$ |
| **WeightedSHAP** | | | | | | | | | | | | | |
| Mean | $1.24 \times 10^{-1}$ | $1.04 \times 10^{-2}$ | $5.98 \times 10^{-2}$ | $1.33 \times 10^{-1}$ | $1.58 \times 10^{-2}$ | $3.36 \times 10^{-2}$ | $7.52 \times 10^{-2}$ | $1.51 \times 10^{-1}$ | $5.43 \times 10^{-2}$ | $4.40 \times 10^{-2}$ | $6.43 \times 10^{-2}$ | $6.64 \times 10^{-2}$ | $6.93 \times 10^{-2}$ |
| 1st Quartile | $1.82 \times 10^{-5}$ | $1.47 \times 10^{-3}$ | $2.45 \times 10^{-3}$ | $4.07 \times 10^{-3}$ | $1.01 \times 10^{-3}$ | $4.46 \times 10^{-3}$ | $1.22 \times 10^{-2}$ | $1.23 \times 10^{-2}$ | $6.90 \times 10^{-3}$ | $3.82 \times 10^{-3}$ | $1.38 \times 10^{-2}$ | $6.67 \times 10^{-3}$ | $5.76 \times 10^{-3}$ |
| 2nd Quartile | $2.94 \times 10^{-4}$ | $3.48 \times 10^{-3}$ | $1.26 \times 10^{-2}$ | $1.20 \times 10^{-2}$ | $4.98 \times 10^{-3}$ | $1.85 \times 10^{-2}$ | $5.33 \times 10^{-2}$ | $2.96 \times 10^{-2}$ | $1.90 \times 10^{-2}$ | $3.07 \times 10^{-2}$ | $3.59 \times 10^{-2}$ | $2.85 \times 10^{-2}$ | $2.07 \times 10^{-2}$ |
| 3rd Quartile | $4.58 \times 10^{-3}$ | $1.42 \times 10^{-2}$ | $3.72 \times 10^{-2}$ | $3.56 \times 10^{-2}$ | $1.74 \times 10^{-2}$ | $4.74 \times 10^{-2}$ | $1.10 \times 10^{-1}$ | $5.53 \times 10^{-2}$ | $7.07 \times 10^{-2}$ | $6.71 \times 10^{-2}$ | $8.94 \times 10^{-2}$ | $9.13 \times 10^{-2}$ | $5.33 \times 10^{-2}$ |

Probabilistic Values: Error vs Sample Complexity (Neural Net Ground Truth)

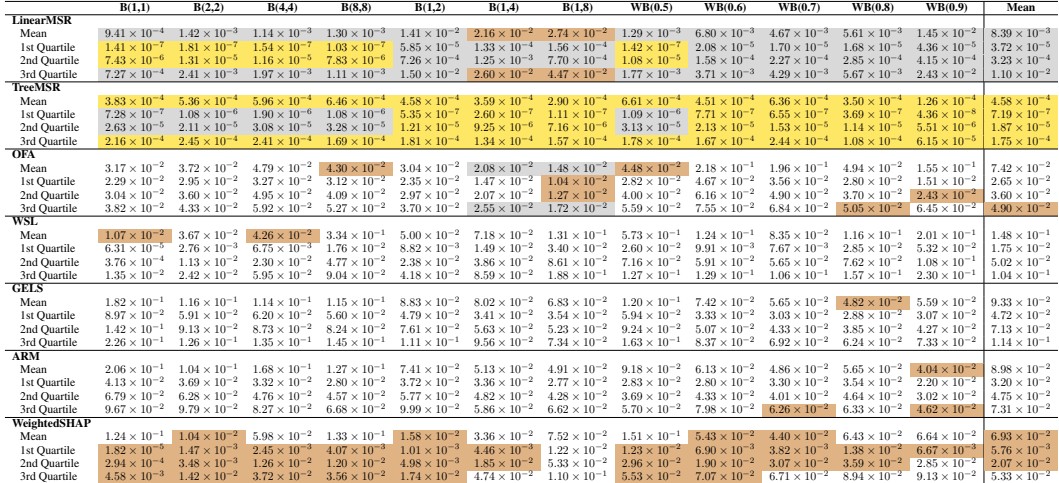

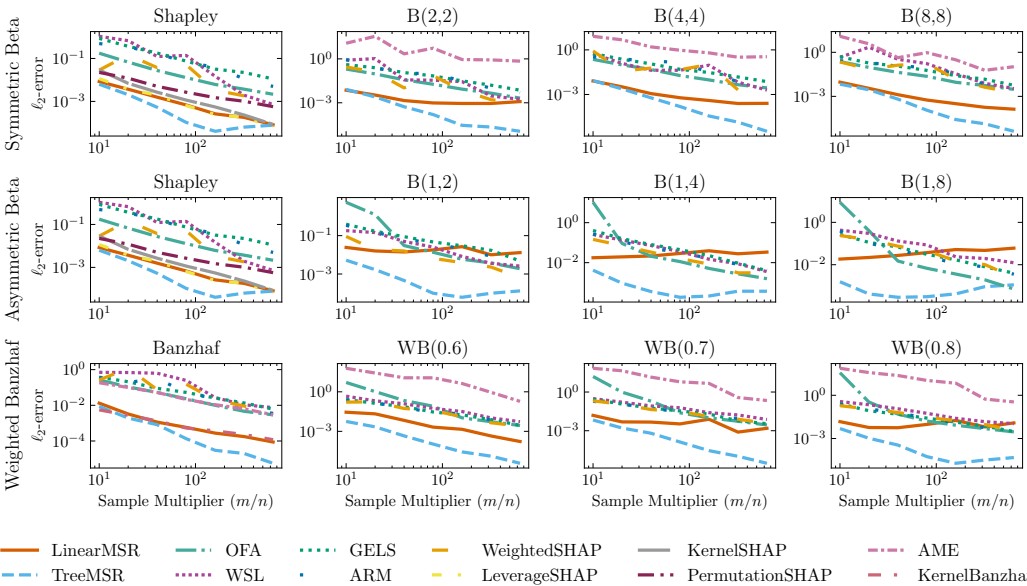

Figure 6: Error between the estimated and true probabilistic values by complexity. Each subplot shows results for a different probabilistic value with the error averaged over all large datasets ($n \geq 30$). The lines report the mean error over 10 runs. Tree and Linear MSR give the best performance, often by several orders of magnitude especially when the number of samples is large.

# F   Experiments by Noise

In many settings, access to the value function is noisy. For example, $v$ may be the expectation over a distribution that is expensive to exactly compute. Instead, we may estimate the expectation, and hence the values we observe are noisy. In this experiment, we add normally distributed noise to the values passed into each estimator. The plots show the performance of each estimator by the magnitude of this noise.

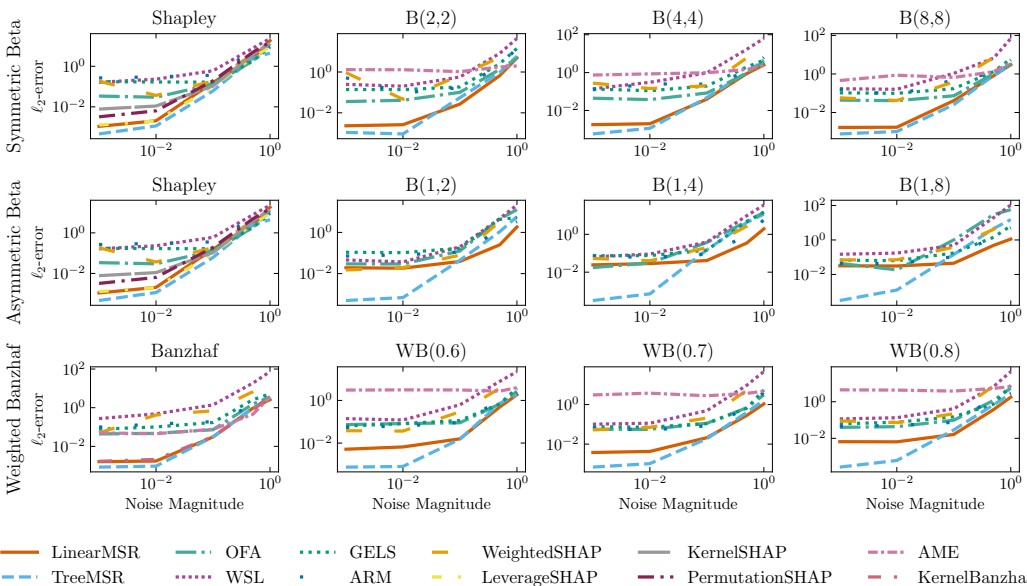

Figure 7: Error between the estimated and true probabilistic values as a function of noise magnitude. Each subplot shows results for a different probabilistic value with the error averaged over all small datasets ($n < 30$). The lines report the mean error over 10 runs. Tree MSR gives the best performance, often by several orders of magnitude especially when the magnitude of the noise is small.

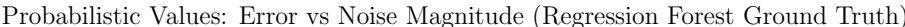

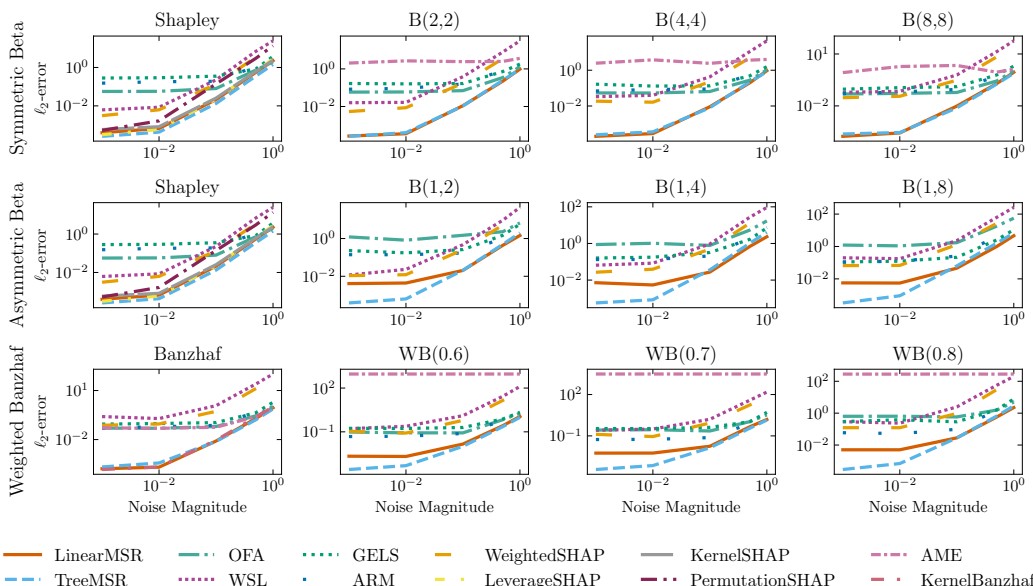

Figure 8: Error between the estimated and true probabilistic values as a function of noise magnitude. Each subplot shows results for a different probabilistic value with the error averaged over all large datasets ($n \geq 30$). The lines report the mean error over 10 runs. Tree MSR gives the best performance, often by several orders of magnitude especially when the magnitude of the noise is small.

# G   Dataset Descriptions

Table 4: A summary of the datasets used in our experiments, including source, access method, license, and number of features $n$.

| Dataset | $n$ | Source / Citation | Access Method | License |
|---------|-----|-------------------|---------------|---------|
| **Adult** | 12 | [Koh96] | `shap.datasets` | CC-BY 4.0 |
| **Forest Fires** | 13 | [CM07] | UCI ML Repo[6] | CC-BY 4.0 |
| **Real Estate** | 15 | [YH09] | UCI ML Repo[7] | CC-BY 4.0 |
| **Bike Sharing** | 16 | [FTG14] | OpenML[8] | Public Domain |
| **Breast Cancer** | 30 | [SWM93] | `sklearn.datasets` | CC-BY 4.0 |
| **Independent** | 60 | [LL17] | `shap.datasets` | MIT |
| **NHANES** | 79 | [Cen23] | `shap.datasets` | Public Domain |
| **Communities** | 101 | [RC02] | `shap.datasets` | CC-BY 4.0 |

---

[6]`https://archive.ics.uci.edu/ml/datasets/forest+fires`
[7]`https://archive.ics.uci.edu/ml/datasets/Real+estate+valuation+data+set`
[8]`https://www.openml.org/d/42712`

