# OpenReview forum: "Regression-adjusted Monte Carlo Estimators for Shapley Values and Probabilistic Values"
_NeurIPS.cc/2025/Conference — NeurIPS 2025 poster_

### Official Review · Reviewer_38gP · 2025-06-23

**Clarity:** 2
**Significance:** 3
**Originality:** 3
**Rating:** 4
**Confidence:** 5

**Summary:**

The paper proposes a technique that combines Monte Carlo sampling with a regression-based formulation. This approach is claimed to reduce the variance of approximation, thereby improving the estimation of probabilistic values. The authors also introduce treeprob, a method for exactly computing all probabilistic values when interventional utility is used. Experimental results support the potential effectiveness of the proposed technique.

**Questions:**

- Why not combine $\mathcal{S}\^{\mathrm{train}}$ and $\mathcal{S}\^{\mathrm{test}}$ for both learning and approximating, given that they are drawn from the same distribution?
- Could the authors provide results of substituting other Monte Carlo baselines in both linearMSR and treeMSR?

**Ethical Concerns:**

["NO or VERY MINOR ethics concerns only"]

**Final Justification:**

Although the presented theory has certain flaws, such as not positioning the proposed MSR estimator relative to existing ones in terms of theoretical convergence rate, and Theorem 2.1 for employing the proposed regression-adjusted trick being clearly naive, I believe the trick itself is promising, and future work may yield improved results both theoretically and empirically. In my view, a better theorectical framework is far beyond what is presented in this work.

**Limitations:**

yes

**Paper Formatting Concerns:**

I did not spot any formatting issues.

**Quality:**

3

**Strengths And Weaknesses:**

**Strengths**
- The proposed approach appears promising for developing variance-reduced Monte Carlo methods for approximating probabilistic values.

**Weaknesses**
#### Major
- Basically, the proposed procedure can be summarized as follows: (1) learning a function $f$ to approximate the given utility function $v$, where $\phi(f)$ is cheap to compute exactly; and then (2) approximate $\phi(v)$ as $\phi(f) + \hat{\phi}(v-f)$. In particular, $\hat{\phi}(v-f)$ is an approximation of $\phi(v-f)$ computed using th Monte Carlo method introduced in Eq. (2). That being said,  $\hat{\phi}(v-f)$ could also be computed using any existing Monte Carlo methods. This suggests that all the considered Monte Carlo baselines could potentially benefit from the same learned function $f$.


- On the other hand, while the proposed Monte Carlo method appears novel, it is unclear whether it achieves faster convergence rate either empirically or theoretically compared to the existing baselines.

All in all, the paper lacks experiments demonstrating that the proposed Monte Carlo method is the most effective when paired with the learned $f$.


#### Minor
- I suggest that the authors include more details on the implementation of treeMSR. After examining the provided code, I found that the features used are binary; the baseline is the zero vector, and the explicand is the one vector. This important information is missing in the paper.

- In the proof of theorem 2.1, the randomness involved in learning $f$ is not accounted. As a result, the stated result applies only to the specific Monte Carlo method defined in Eq. (3). The current statement of theorem 2.1 may therefore be misleading.


**Typos**
- According to Eq. (7), the first equality in Eq. (8) should be changed to $\leq$.
- In line 26 of Algorithm 2, it should be $\phi[\mathrm{node.feat}] \gets \phi[\mathrm{node.feat}] + (\mathrm{pos}\_{e} + \mathrm{neg}\_{b})$.

---

> ### Author Rebuttal · Authors · 2025-07-30
>
> > all the considered Monte Carlo baselines could potentially benefit from the same learned function $f$.
>
> We agree with this point, and actually view it as a *strength* rather than a *weakness* of our approach. We selected MSR as our “baseline” estimator because, in addition to being simple to implement, it is
>
> 1) unbiased (contrasting with Kernel SHAP and Leverage SHAP), and
>
> 2) computationally efficient in the sense that every function evaluation $v(S)$ is used to update every estimate (contrasting with Permutation SHAP and MC).
>
> As far as we know, MSR uniquely satisfies these two properties among probabilistic value estimators.
>
> > the paper lacks experiments demonstrating that the proposed Monte Carlo method is the most effective when paired with the learned $f$.
>
> Since the main contribution of our work is the regression adjustment technique (learning an $f$ for which we can compute probabilistic values exactly), considering a wide variety of alternative baseline estimators is outside the scope of the paper. That being said, we just ran some experiments with MC as the baseline; please see below.
>
> > On the other hand, while the proposed Monte Carlo method appears novel, it is unclear whether it achieves faster convergence rate either empirically or theoretically compared to the existing baselines.
>
> For fixed sample budgets, Tree MSR achieves an average error that is $1.75\times$ lower than the prior best estimator, Leverage SHAP, across eight popular datasets (Table 1). As for convergence rate, we explore how accurately the estimators recover the true Shapley values (Figure 2) and probabilistic values (Figure 3) as a function of sample size. From these figures, we conclude that, for most datasets, *Tree MSR does gives faster convergence empirically*. We justify this finding theoretically in Theorem 2.1, where we show the performance of our estimator depends on how well the (randomly) learned $f$ fits $v$.
>
> > I suggest that the authors include more details on the implementation of treeMSR. After examining the provided code, I found that the features used are binary; the baseline is the zero vector, and the explicand is the one vector. This important information is missing in the paper.
>
> We will add more details! One note: In the interventional setting where we apply Tree Prob, a point is fully specified by the binary indicator of which features come from the “explicand” and which come from the “baseline”. Since the tree-based model can learn in a scale-invariant and shift-invariant way with respect to these features, it actually suffices to only consider binary inputs. We make this clear in the updated version of the paper.
>
> > In the proof of theorem 2.1, the randomness involved in learning $f$ is not accounted. As a result, the stated result applies only to the specific Monte Carlo method defined in Eq. (3). The current statement of theorem 2.1 may therefore be misleading.
>
> We can add a comment reminding the reader that f itself is a random function. Since independent samples are used to fit $f$ (line 3-4 in Algorithm 1) and to approximate Shapley values (line 6-7) the statement of Theorem 2.1 is true regardless of the outcome of f.
>
> > Typos
>
> Thank you!
>
> > Why not combine $\mathcal{S}^\text{train}$ and $\mathcal{S}^\text{test}$ for both learning and approximating, given that they are drawn from the same distribution?
>
> Good question! Our variance analysis in the proof of Theorem 2.1 relies on independence between the final estimate and the learned function $f$. We use $\mathcal{S}^\text{train}$ to learn the function $f$, and then $\mathcal{S}^\text{test}$ for the samples used to build the final estimate.
>
> > Could the authors provide results of substituting other Monte Carlo baselines in both linearMSR and treeMSR?
>
> Yes! We compare several Shapley value estimators in the small sample regime $m=10n$ on the four small datasets below.
>
> ### Dataset: Adult (n=12)
>
> | Estimator | Mean | 1st | 2nd | 3rd | Max |
> |--------------------|-----------|-----------|-----------|-----------|-----------|
> | LinearMC | 6.80e-01 | 2.70e-02 | 1.04e-01 | 1.31e-01 | 2.53e+00 |
> | TreeMC | 1.30e-05 | 2.92e-08 | 5.40e-08 | 1.34e-07 | 1.22e-04 |
> | LinearMSR | 1.59e-04 | 1.77e-05 | 4.43e-05 | 5.07e-05 | 4.89e-04 |
> | TreeMSR | 1.09e-06 | 3.61e-08 | 4.19e-08 | 1.68e-07 | 3.12e-06 |
> | KernelSHAP | 5.19e-04 | 1.01e-04 | 1.18e-04 | 1.90e-04 | 1.15e-03 |
> | LeverageSHAP | 1.87e-04 | 1.13e-05 | 3.06e-05 | 4.64e-05 | 5.34e-04 |
> | PermutationSHAP | 4.83e-03 | 2.65e-04 | 4.14e-04 | 5.85e-04 | 2.04e-02 |
>
> ### Dataset: Forest Fires (n=13)
>
> | Estimator | Mean | 1st | 2nd | 3rd | Max |
> |--------------------|-----------|-----------|-----------|-----------|-----------|
> | LinearMC | 4.64e-01 | 3.47e-02 | 1.03e-01 | 1.09e-01 | 1.99e+00 |
> | TreeMC | 1.29e-05 | 2.54e-07 | 1.18e-06 | 2.09e-06 | 4.15e-05 |
> | LinearMSR | 6.41e-05 | 1.27e-05 | 1.50e-05 | 1.54e-05 | 2.93e-04 |
> | TreeMSR | 1.21e-06 | 6.79e-08 | 1.00e-07 | 1.05e-07 | 8.51e-06 |
> | KernelSHAP | 6.72e-05 | 7.45e-06 | 1.51e-05 | 1.89e-05 | 1.86e-04 |
> | LeverageSHAP | 5.78e-05 | 6.83e-06 | 8.97e-06 | 1.86e-05 | 1.64e-04 |
> | PermutationSHAP | 7.63e-04 | 1.40e-04 | 1.42e-04 | 1.70e-04 | 2.53e-03 |
>
> ### Dataset: Real Estate (n=15)
>
> | Estimator | Mean | 1st | 2nd | 3rd | Max |
> |--------------------|-----------|-----------|-----------|-----------|-----------|
> | LinearMC | 7.30e+00 | 2.27e-01 | 4.37e-01 | 6.58e-01 | 6.20e+01 |
> | TreeMC | 1.21e-10 | 4.54e-12 | 4.83e-12 | 1.05e-11 | 8.63e-10 |
> | LinearMSR | 5.94e-06 | 3.21e-07 | 5.74e-07 | 7.47e-07 | 2.44e-05 |
> | TreeMSR | 1.95e-10 | 4.19e-12 | 4.63e-12 | 7.32e-12 | 1.60e-09 |
> | KernelSHAP | 8.02e-06 | 3.00e-07 | 4.86e-07 | 5.28e-07 | 4.87e-05 |
> | LeverageSHAP | 8.77e-06 | 2.38e-07 | 4.20e-07 | 1.18e-06 | 3.79e-05 |
> | PermutationSHAP | 3.66e-05 | 4.53e-08 | 4.29e-07 | 1.22e-06 | 2.03e-04 |
>
> ### Dataset: Bike Sharing (n=16)
>
> | Estimator | Mean | 1st | 2nd | 3rd | Max |
> |--------------------|-----------|-----------|-----------|-----------|-----------|
> | LinearMC | 2.18e+00 | 4.83e-02 | 1.08e-01 | 2.63e-01 | 1.74e+01 |
> | TreeMC | 2.05e-06 | 1.31e-08 | 1.52e-08 | 1.99e-07 | 8.10e-06 |
> | LinearMSR | 5.82e-05 | 1.13e-06 | 5.11e-06 | 9.26e-06 | 1.89e-04 |
> | TreeMSR | 2.96e-07 | 4.78e-10 | 5.78e-10 | 1.60e-08 | 1.96e-06 |
> | KernelSHAP | 7.10e-05 | 1.18e-06 | 1.73e-05 | 2.01e-05 | 1.61e-04 |
> | LeverageSHAP | 5.19e-05 | 3.73e-06 | 5.26e-06 | 9.99e-06 | 1.28e-04 |
> | PermutationSHAP | 2.16e-04 | 1.47e-05 | 2.94e-05 | 4.82e-05 | 3.82e-04 |
>
> While Tree MC is quite good, we generally find that Tree MSR gives error about an order of magnitude smaller. We believe this is due to the differences in the final regression-adjusted estimate: each sample is used for every estimate in MSR, whereas each sample is only used for one estimate in MC.

---

> > ### Comment · Reviewer_38gP · 2025-08-04
> >
> > Thank you to the authors for taking the time to respond. In my opinion, the proposed regression-adjusted trick is promising, as it is fully complementary to existing Monte Carlo methods and serves as a bridge between what can be exactly computed and what can only be approximated. However, what prevented me from giving a higher score is that I find the theoretical justification somewhat lacking, although this could be addressed in future work. I will make my decision later.

---

### Official Review · Reviewer_BiBE · 2025-07-01

**Clarity:** 3
**Significance:** 4
**Originality:** 4
**Rating:** 5
**Confidence:** 5

**Summary:**

This paper proposes using the control variates (CV) method or regression sampling (RS) method for computing the Shapley value and its variants to address the well-known computational challenges. While CV and RS are well-established techniques in Monte Carlo estimation, to the best of my knowledge, this is the first proposal to apply these methods to Shapley value calculation, which is a noteworthy contribution.

As demonstrated in general Monte Carlo contexts, both CV and RS are undoubtedly effective. For Shapley value computation, the baseline function can be chosen as a linear function, since in that case the exponential sum required for the Shapley value can be computed exactly.

The idea is straightforward. Since existing sampling approaches such as LeverageSHAP are already highly optimized, direct performance comparison may be challenging. Nevertheless, the authors’ proposal is meaningful as an additional acceleration method for SHAP, complementing LeverageSHAP. I support acceptance of this paper.

**Questions:**

I found Fig. 1(b) confusing. The Correlated and Community data points do not appear to satisfy the unbiasedness property. Why do they seem so biased? In theory, the estimated values should be distributed both above and below the true value, so that the deviations cancel out on average. Why is this not observed in the figure?

**Ethical Concerns:**

["NO or VERY MINOR ethics concerns only"]

**Limitations:**

CV and RS is well-known. Contribution can be seen incremental from a general Monte Carlo study perspective. However, application to Shapley value computation is novel enough. This would should be accepted.

**Paper Formatting Concerns:**

None.

**Quality:**

4

**Strengths And Weaknesses:**

Strength:
- The first introduction of the CV or RS method to Shapley values.
- Demonstrated variance reduction.

Weakness:
- Depending on how the performance is measured, improvement may not be drastic as compared to highly optimized alternatives.

---

> ### Author Rebuttal · Authors · 2025-07-30
>
> Thank you for your positive review!
>
> > Depending on how the performance is measured, improvement may not be drastic as compared to highly optimized alternatives.
>
> While it doesn’t give SOTA in every setting, our Tree MSR achieves an average error $1.75 \times$ lower than the prior best Shapley value estimator, Leverage SHAP, on eight common datasets.
>
> > I found Fig. 1(b) confusing. The Correlated and Community data points do not appear to satisfy the unbiasedness property. Why do they seem so biased? In theory, the estimated values should be distributed both above and below the true value, so that the deviations cancel out on average. Why is this not observed in the figure?
>
> Figure 1(b) shows the true Shapley values vs. estimates produced by the **standard MSR estimator on one random run**. The multiple points for each dataset (e.g. for Correlated and Community) correspond to different features in the dataset. While the estimated Shapley value for each individual feature is unbiased, there is substantial correlation between the estimates for different features, leading to this clustering. The correlation is due to the fact that every function evaluation $v(S)$ is used to estimate the Shapley value for *every* feature. Large values of $v(S)$ in particular can have substantial impact. To observe unbiasedness, we would need to re-run the same experiment with different random seeds. If we do so, we see that the estimates for Correlated and Communities do not consistently appear on one side of the diagonal line.
>
> While the computational advantage of MSR is its “maximum reuse”, this property comes with a cost in variance and correlation. We view Regression MSR as a fix: The variance of Regression MSR depends on $[ f(S) \approx v(S) ]^2$ rather than $[v(S)]^2$ like standard MSR. So, even if some $v(S)$ is large, we can still get remarkably accurate estimators provided $f(S) \approx v(S)$.

---

### Official Review · Reviewer_Q2YG · 2025-07-02

**Clarity:** 3
**Significance:** 3
**Originality:** 3
**Rating:** 5
**Confidence:** 3

**Summary:**

This paper proposed a novel estimator for general probabilistic values such as Shapley values and Banzhaf values. The authors combined Monte Carlo sampling with regression-based methods to construct an unbiased estimator. Specifically, the proposed method used regression as a variance reduction technique for the Maximum Sample Reuse method, thereby integrating the strengths of both approaches.

**Questions:**

See weaknesses

**Ethical Concerns:**

["NO or VERY MINOR ethics concerns only"]

**Final Justification:**

I appreciate the strengths of this manuscript and the authors’ detailed responses. I am inclined to support its acceptance.

**Limitations:**

yes

**Quality:**

3

**Strengths And Weaknesses:**

**Strengths:**

1.	The proposed estimator has several desirable properties: (1) it is unbiased, (2) it is applicable to any probabilistic value (e.g., Shapley value and Banzhaf value), (3) it allows the use of any learned function $f$  to approximate the value function $v$, and (4) the variance of the estimator depends on the quality of the surrogate function $f$. As such, when $f$ is accurate, the variance can be significantly reduced.

2.	The authors analyze the error bound of the estimator under an arbitrary sampling distribution $D$ , which improves the theoretical soundness and generality of the method.

3.	The authors implement the estimator using both linear and tree-based models for the surrogate function $f$, with the tree-based models exhibiting lower approximation error in practice.

**Weaknesses:**

1.	It would be helpful if the authors could provide a more detailed derivation of Equation (5) in the appendix, including a formal proof of its unbiasedness and an analysis of its variance.



2.	Since the variance of the proposed estimator depends on the surrogate function $f$, it would be beneficial to provide a more detailed discussion of how different choices of $𝑓$ affect the variance and estimation accuracy. For example, to what extent does the use of linear versus non-linear models influence the quality of the estimated probabilistic values? Could shallow neural networks serve as effective approximators in this context? A more detailed analysis of how the choice of $f$ impacts estimation error would strengthen the paper.



3.	It is unclear how the computational complexity of the proposed method compares to existing approaches. In particular, it would be helpful to clarify whether the estimator has similar time complexity to the Maximum Sample Reuse method. A discussion or empirical comparison of the time complexity (or runtime) would improve the clarity of the contribution.

4.  The authors could add a summary table in the appendix comparing the proposed method with existing baselines (e.g., in terms of unbiasedness, variance, computational cost, use of surrogate models, and other relevant properties), which would help readers better understand the advantages and limitations of each method.

---

> ### Author Rebuttal · Authors · 2025-07-30
>
> Thank you for your detailed comments! We respond below:
>
> > It would be helpful if the authors could provide a more detailed derivation of Equation (5) in the appendix, including a formal proof of its unbiasedness and an analysis of its variance.
>
> These derivations are provided in proof of Theorem 2.1 (Appendix A). We show that $\tilde{\phi}_i$ is unbiased in the equation under Line 370, and analyze its variance in Equation (7). In response to your comment, and that of Reviewer z7xn, we will make this analysis clear in the *main body* of the updated paper.
>
> > to what extent does the use of linear versus non-linear models influence the quality of the estimated probabilistic values? Could shallow neural networks serve as effective approximators in this context? A more detailed analysis of how the choice of impacts estimation error would strengthen the paper.
>
> A crucial component of our proposed Regression MSR estimator is that the probabilistic values of the approximation $f$ need to be efficiently computable. (Otherwise, we would be stuck with the initial problem of estimating the probabilistic values of $f$.)
>
> We are aware of two primary function classes with this property:
>
> * The coefficients of **linear functions** are always their probabilistic values. (This can directly be seen directly in Equation (1) with the *probabilistic* constraint that $\sum_{\ell=0}^{n-1} {n-1 \choose ell} p_\ell = 1$.)
>
> * We show in Appendix C how the probabilistic values of (multiple) **decision trees** can be computed efficiently. A special case of this was known for Shapley values, and basically follows from the “case”-like structure of trees.
>
> The issue with neural networks is that computing their Shapley values is generally slow, requiring O(2^n) time even when the network is shallow or has few parameters.
>
> > it would be helpful to clarify whether the estimator has similar time complexity to the Maximum Sample Reuse method. A discussion or empirical comparison of the time complexity (or runtime) would improve the clarity of the contribution.
>
> Great question! In our experience, the dominating cost is computing $m$ function evaluations of $v$. Here’s one way to see why: If $v$ is as expressive as a shallow neural network, this time complexity is already $O(m n^2)$ and likely much higher. (A single fully connected layer between $n$ nodes requires $O(n^2)$ operations, repeated for each of the $m$ different $n$-dimensional inputs.)
>
> Beyond the time to get evaluations of $v$, the additional computational step of Regression MSR is fitting the function $f$:
>
> * When $f$ is linear, solving a linear system with $m$ equations and $n$ rows takes $O(mn^2)$ time. In practice, this is very fast due to excellent linear algebra libraries. Note: Kernel SHAP and Leverage SHAP have this same time complexity because they are also solving a linear system of the same size.
>
> * When $f$ is tree-based model, fitting the function requires roughly $O(T * m * n)$ time where $T$ is the number of trees. In practice, we use the default $T=100$ for XGBoost. Due to its extensive optimizations, fitting XGboost is remarkably fast in practice.
>
> > The authors could add a summary table in the appendix comparing the proposed method with existing baselines (e.g., in terms of unbiasedness, variance, computational cost, use of surrogate models, and other relevant properties), which would help readers better understand the advantages and limitations of each method.
>
> Great idea! We agree a summary table would be useful for the community. Please see below for an initial Shapley value estimator summary. We will continue to update, adding more estimators and properties, for the final version of the paper. Let $T_m$ be the time to evaluate the value function $v$ on $m$ different subsets.
>
> | Method             | Bias     | Time Complexity       | Regression-based       |
> |--------------------|----------|------------------------|-------------------------|
> | Kernel SHAP        | Biased   | $T_m + O(mn^2)$        | Linear Model            |
> | Leverage SHAP      | Biased   | $T_m + O(mn^2)$        | Linear Model            |
> | Permutation SHAP   | Unbiased | $T_m + O(m + n)$       | No                      |
> | MC                 | Unbiased | $T_m + O(m + n)$       | No                      |
> | MSR                | Unbiased | $T_m + O(m + n)$       | No                      |
> | Regression MSR     | Unbiased | $T_m + O(mn^2)$        | Linear or Tree Model    |

---

### Official Review · Reviewer_z7xn · 2025-07-03

**Clarity:** 2
**Significance:** 3
**Originality:** 3
**Rating:** 5
**Confidence:** 3

**Summary:**

The article addresses the estimation of probabilistic values such as Shapley values and Banzhaf values without the need to perform $2^{n}$ computations and accesses to the payoff function.
They propose using a simpler function, $f(S): 2^{[n]} \to \mathcal{R}$, to approximate $v(S)$. This function is then used to exploit the linearity property of probabilistic values to find a formulation that can estimate the contributions consistently, which is proven theoretically.
The proposed algorithm assumes that $\phi(f)$ can be computed fast, i.e. $f$ is either linear or tree-based.
The article also provides intensive numerical experiments analyzing the sample complexity of the proposed algorithm and comparing it to the many published methods.

**Questions:**

1) Equation (3) seems central to the proposed algorithm, but no intuition is provided. It’s also unclear which cited work this equation comes from. Can you clarify clarify its origin and provide some explanation?

2) Theorem 2.1 is the main contribution of the article, I think it can be stated better, although equation (6) does entail consistency results, I believe the main contribution of the article can be more readily accessible if consistency ($E[\hat{\phi}_{i}] = \phi_{i}$) and variance results are mentioned explicitly in the main body of the article.

3) is the division between Monte Carlo and regression-based methods strictly correct? For instance, doesn’t KernelSHAP also involve Monte Carlo sampling, not just regression?

4) Some explanations are hard to follow. For example, in line 35, the first sentence appears without context, and the topic shifts abruptly right after. This is also seen in sections 1.1

5) Key related works like Mitchell et al., 2022 ("Sampling Permutations for Shapley Value Estimation") and Jethani et al., 2022 ("FastSHAP") are not mentioned. Even though FastSHAP is not based on Monte Carlo, it is relevant. Also, Mitchell et al. show that Owen's halves can be effective, yet this article only discusses Owen’s method in the context of KernelSHAP. I understand that the relevant literature is vast but it does draw attention that the method is not compared to such prevalent works

**Ethical Concerns:**

["NO or VERY MINOR ethics concerns only"]

**Final Justification:**

I believe the results of the article are significant. I had concerns about the comparisons and some points in the articles. The authors have convinced me on all points. I believe the article is interesting to people working in the area of XAI and is technically solid with impact on the area.

**Limitations:**

yes

**Quality:**

3

**Strengths And Weaknesses:**

Strengths:
1) The article addresses an important problem in XAI which is the efficient computation of Shapley values
2) It provides sound theoretical guarantees.
3) The experimental results are thorough and convincingly demonstrate the method’s effectiveness.

Weaknesses:
1) The presentation lacks clarity in key areas, including both the preliminary literature overview, the discussion of closely related methods, and even the theoretical contribution
2) The manuscript omits discussion of several relevant works in the literature on fast Shapley value estimation. While the literature is admittedly large, a more comprehensive review would strengthen the article.

---

> ### Author Rebuttal · Authors · 2025-07-30
>
> Thank you for your careful reading! We appreciate your clarifying questions, and pointers to specific related works. We respond in detail below:
>
> > Equation (3) seems central to the proposed algorithm [...] Can you clarify its origin and provide some explanation?
>
> The Maximum Sample Reuse (MSR) estimator described in Equation (3) was originally proposed to estimate Banzhaf values [WJ23]. Since then, it has been generalized to other probabilistic values, albeit with differing notations [KBMH24, LY24a, LY24b]. The key insight is that we can write the probabilistic values as
>
> $$
> \phi_i (v) = \sum_{S \subseteq [n] \setminus \{ i \}} p_{|S|} [ v(S \cup \{i \}) - v(S) ]
> $$
> $$
> = \sum_{S  \subseteq [n] : i \in S} p_{|S| -1} v(S) - \sum_{S \subseteq [n]: i \notin S} p_{|S|} v(S)
> $$
> $$
> = \sum_{S \subseteq [n]} v(S) [p_{|S|-1} {1}[i \in S] - p_{|S|} {1}[i \notin S] ]
> $$
>
> Compared to the first equation, the final equation makes it clear how $v(S)$ appears (albeit with a potentially different sign and weighting) in every probabilistic value. This observation naturally suggests the MSR estimator in Equation (3), where each $S$ is sampled, and reweighted, so that the estimator is right in expectation. We will update the paper to make this motivation clear.
>
> > I believe the main contribution of the article can be more readily accessible if consistency ($E[\hat{\phi}{i}] = \phi{i}$) and variance results are mentioned explicitly in the main body of the article.
>
> We think this is a good suggestion, and will plan on including the consistency and variance directly in Theorem 2.1.
>
> > is the division between Monte Carlo and regression-based methods strictly correct? For instance, doesn’t KernelSHAP also involve Monte Carlo sampling, not just regression?
>
> Good question! In our work, we distinguish between“Monte Carlo” methods that *directly* estimate the linear sum of probabilistic values, and “regression-based” methods that fit an approximation $f$ to the underlying game $v$, then compute the probabilistic values of $f$. While methods like KernelSHAP use *random sampling* to fit a linear function $f$ to $v$, the method ultimately computes $f$ via least squares regression and returns the Shapley values of $f$. (Since $f$ is a linear function, its Shapley values are simply the coefficients for each feature.) We will clarify this distinction in the updated paper..
>
> > in line 35, the first sentence appears without context, and the topic shifts abruptly right after. This is also seen in sections 1.1
>
> Thank you for pointing out these writing discontinuities. We have smoothed out the transitions (Line 35 and Section 1.1 more broadly) in the new version.
>
> > Even though FastSHAP is not based on Monte Carlo, it is relevant. Also, Mitchell et al. show that Owen's halves can be effective, yet this article only discusses Owen’s method in the context of KernelSHAP. I understand that the relevant literature is vast but it does draw attention that the method is not compared to such prevalent works
>
> Due to their widespread applications in interpretability, there is certainly a vast literature on estimating Shapley values, and probabilistic values more broadly. We will plan on expanding our discussion of prior work in the updated paper. We briefly discuss two methods mentioned by the reviewer here:
>
> Fast SHAP is a method for learning a function $f$ that simultaneously fits *multiple* related games $v$. While it has a high overhead in sample complexity to learn $f$, Fast SHAP can give improved results when we want to estimate Shapley values for multiple, similar games simultaneously. In our setting, we are interested in estimating the Shapley values of a *single* game $v$. Due to these incomparable settings, we do not compare to Fast SHAP in our work, but have now highlighted the estimator in the updated related work section in response to your comment.
>
> Owen’s and Owen’s halves are two Shapley value estimators. They both exploit a connection between Shapley values and the expectation of a continuous process. The “halves” augmentation is a kind of paired sampling (sometimes called “antithetical” sampling) that boosts performance. We chose not to compare to Owen’s or Owen’s halves estimators because they were shown in “Sampling Permutations for Shapley Value Estimation” to consistently perform worse than Permutation SHAP (what they call “MC-antithetic”) see e.g., their Figure 9. In our work, we find that Tree MSR already achieves an average $6\times$ improvement over Permutation SHAP. Nevertheless, we will plan on adding discussion of this prior work.

---

> > ### Comment · Reviewer_z7xn · 2025-08-04
> >
> > I thank the authors for their detailed response. They have clarified and answered all my questions and concerns. I will raise my score to an accept.

---

### Decision · Program_Chairs · 2025-09-17

**Decision:**

Accept (poster)

**Comment:**

The paper presents a novel method for efficiently estimating probabilistic values, such as Shapley and Banzhaf values, by combining Monte Carlo sampling with flexible function-based regression. This approach achieves state-of-the-art accuracy across multiple datasets, significantly reducing estimation error compared to existing methods while maintaining unbiased probabilistic value computations.

Strengths of the paper include addressing efficient and accurate computation of Shapley and other probabilistic values with strong theoretical guarantees. The estimator is unbiased, flexible, applicable to any probabilistic value, and achieves variance reduction with accurate surrogates. Experiments are thorough and demonstrate state-of-the-art performance, particularly using tree-based models, and the method introduces novel variance-reduction techniques.

Reviewers raised several concerns, many of which were addressed during the rebuttal and discussion period, including unclear presentation of the literature review, related methods, and theoretical contributions, missing discussion of relevant prior work, and insufficient derivations or proofs. The impact of different surrogate functions on variance and accuracy is underexplored, computational complexity and runtime comparisons are lacking, and it is unclear if the method converges faster than existing Monte Carlo baselines.

The reviewers agree that the paper is technically solid, addresses an important problem in XAI, and is of interest to the community. While some theoretical aspects, such as the convergence analysis and Theorem 2.1, are limited or simplistic, the proposed regression-adjusted approach is promising and may inspire future theoretical and empirical improvements. Overall, the reviewers support acceptance.